## PERSPECTIVE

# The interplay between membrane topology and mechanical forces in regulating T cell receptor activity

Mohammad Ameen Al-Aghbar [1,2], Ashwin K. Jainarayanan [3,4], Michael L. Dustin [4✉] & Steve R. Roffler [1,5✉]

T cells are critically important for host defense against infections. T cell activation is specific because signal initiation requires T cell receptor (TCR) recognition of foreign antigen peptides presented by major histocompatibility complexes (pMHC) on antigen presenting cells (APCs). Recent advances reveal that the TCR acts as a mechanoreceptor, but it remains unclear how pMHC/TCR engagement generates mechanical forces that are converted to intracellular signals. Here we propose a TCR Bending Mechanosignal (TBM) model, in which local bending of the T cell membrane on the nanometer scale allows sustained contact of relatively small pMHC/TCR complexes interspersed among large surface receptors and adhesion molecules on the opposing surfaces of T cells and APCs. Localized T cell membrane bending is suggested to increase accessibility of TCR signaling domains to phosphorylation, facilitate selective recognition of agonists that form catch bonds, and reduce noise signals associated with slip bonds.

Immunity against foreign materials, viruses, pathogens, and cancer relies on a set of specialized immune cells. T cells are key players in adaptive immunity, and naïve T cells typically express ~$1 \times 10^5$ monotypic T-cell receptor (TCR) complexes[1]. Upon TCR ligation with a cognate pMHC molecule (MHC loaded with peptide) on an antigen-presenting cell (APC), immunoreceptor tyrosine-based activation motifs (ITAMs) present in the cytoplasmic tails of the multichain TCR complex are phosphorylated by Lck (lymphocyte-specific protein tyrosine kinase) and Fyn (proto-oncogene tyrosine-protein kinase) members of the Src family of protein tyrosine kinases (PTKs)[2,3]. The phosphorylated ITAMs serve as docking sites for the protein tyrosine kinase Zap70 (zeta chain-associated 70 kDa phosphoprotein), which in turn is activated by Lck phosphorylation (pZap70). Activated Zap70 then phosphorylates other proteins including LAT (linker for activation of T cells), SLP76 (SRC homology 2 (SH2)-domain-containing leukocyte protein of 76 kDa) and PLC-γ1 (phospholipase Cγ1)[4,5]. TCR triggering is followed by the rapid flux of $Ca^{2+}$ from the endoplasmic reticulum[6], which in turn activates PKC (protein kinase C) and Ca-dependent serine/threonine phosphatase calcineurin that dephosphorylates the transcription factor NFAT (nuclear factor of activated T cells) and promotes its rapid translocation to the nucleus[7]. In addition, other signaling pathways are activated such as RAS, NF-κB (nuclear factor kappa B), and MAPK (mitogen-activated protein kinase). As a result, activated T cells begin proliferating and secreting cytokines[8]. Although many models propose how pMHC/TCR ligation initiates signaling, the mechanism remains controversial[9–11]. It has been proposed that the TCR is inherently a mechanosensor[12–18], but the actual form of

[1] Institute of Biomedical Sciences, Academia Sinica, Taipei, Taiwan. [2] Department of Translational Medicine, Sidra Medicine, Doha, Qatar. [3] Interdisciplinary Bioscience Doctoral Training Program and Exeter College, University of Oxford, Oxford, UK. [4] Kennedy Institute of Rheumatology, University of Oxford, Oxford, UK. [5] Graduate Institute of Medicine, College of Medicine, Kaohsiung Medical University, Kaohsiung, Taiwan.
✉email: michael.dustin@kennedy.ox.ac.uk; sroff@ibms.sinica.edu.tw

force that is required to trigger TCR signaling is still undetermined[19]. Here, we put forward a model where membrane bending driven by passive and active mechanisms mediates TCR triggering.

## Organization of membrane proteins involved in T cell activation

The structure of interacting receptors and repellers sets the stage for passive mechanisms involved in TCR triggering. Thus, we first introduce the structure of a core of relevant receptors and repellers and how this leads to the organization on different length scales.

The TCR receptor complex is composed of TCRα and β chains and six associated non-polymorphic chains: CD3γ-CD3ε and CD3δ-CD3ε heterodimers, and a CD3ζζ homodimer (also referred as ζζ or CD247) that are arranged in defined orientations with respect to the TCR[20,21]. TCRαβ extends 7.5 nm from the T cell surface, while the transmembrane domains of TCR and CD3

proteins span 4 nm in depth and 3.5 nm in width in the supported lipid bilayer (SLB)[22] (Fig. 1a).

The T cell surface is rich with co-receptors that play fundamental roles in T cell activation. CD4 consists of 4 Ig-like domains tilted from the cell surface at an angle of 70°[23]. CD8 is a dimeric glycoprotein of CD8α and β chains, each composed of an Ig-like domain connected by a long stalk to a transmembrane domain[24]. CD4 and CD8 have low affinities for MHC-II ($K_d$ ~200 μM)[25,26] and MHC-I ($K_d$ ~50–150 μM)[24,27,28], respectively. Therefore, pMHC/TCR ligation helps recruit Lck that is associated with CD4 and CD8 cytoplasmic tails to CD3 ITAMs[29,30]. CD28 is another important co-receptor that is expressed as a disulfide-linked dimer. Ligation of CD28 to CD80 or CD86 on the surface of APCs provides crucial co-stimulation for signal amplification[31].

T cells also express many important adhesion molecules that enhance cell adhesion and amplify cytokine production[32]. CD2 has 2 Ig-like extracellular domains and one long proline-rich cytoplasmic tail[33]. CD58 and CD48 on APCs serve as ligands to

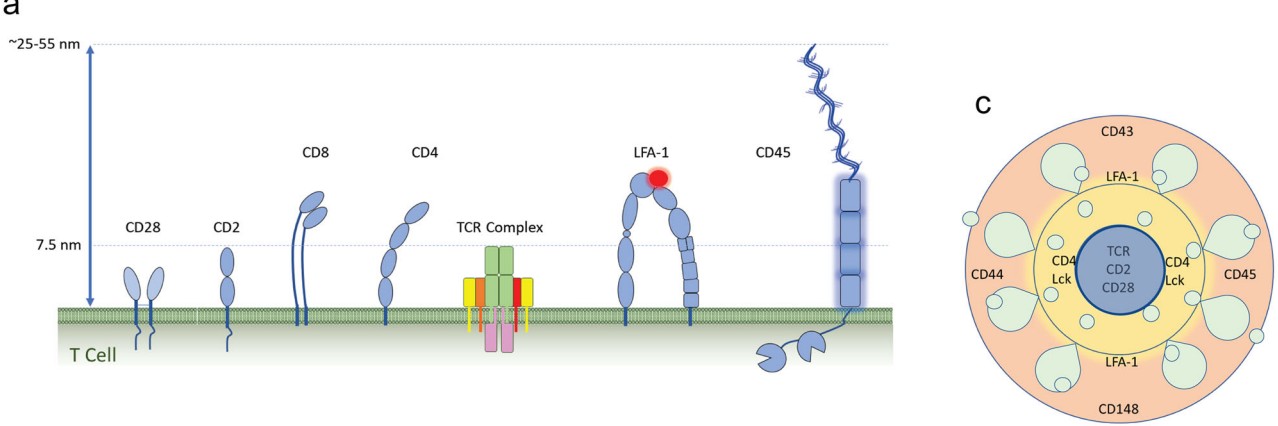

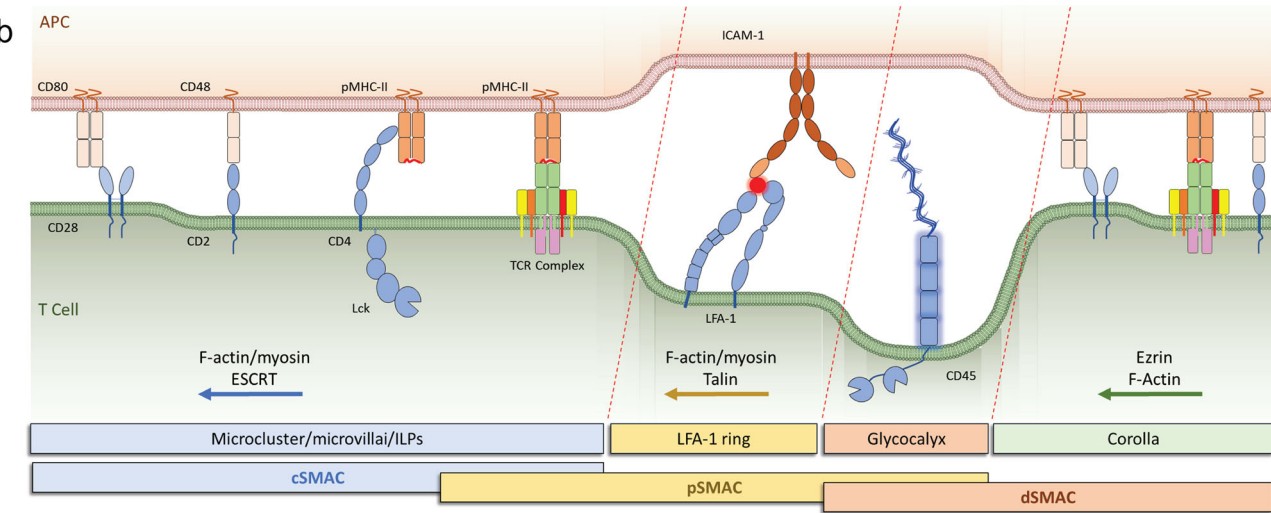

**Fig. 1 T cell surface and the immune synapse. a** The T cell surface is populated with receptors displaying different dimensions ranging from one Ig-like domain (3.5 nm) to those with long and bulky extracellular domains such as CD45 with a rigid core of 15.2 nm and variable mucin-line domain that extends the height to 10 (CD45R0), 20 (CD45RA), or 40 (CD45RABC) nm. The repetition of the mucin-like domain may increase the space it occupies in cell interfaces. **b** Upon APC/T cell engagement, and following TCR triggering, T cell surface proteins sort based on size and F-actin based transport to distinct zones. The mature immune synapse takes 10–30 min to develop, and three zones are arranged in circles, with proteins subjected to strong F-actin and ESCRT based transport in the cSMAC, proteins linked to F-actin/myosin, such as LFA-1 ligated to ICAM-1, in the pSMAC, and proteins with the weakest F-actin coupling in the dSMAC. **c** In addition, some proteins linked directly to TCR triggering such as Lck kinase accumulate in the pSMAC where TCR signaling is initiated. The CD2–CD58 interaction forms a peripheral corolla of close interactions in the dSMAC when a sufficient number of receptors are engaged. TCR microclusters form at the outer edge, translocate through the corolla and pSMAC to reach the cSMAC.

CD2 on human and rodent T cells, respectively[34]. LFA-1 (lymphocyte function-associated antigen 1), one of the most important adhesion proteins in T cells, is a heterodimer of integrin αL (CD11a) and β2 (CD18). LFA-1 has three different conformations based on its activation status: a bent inactive form, a low-affinity form (closed headpiece), and a high-affinity form (open headpiece)[35]. ICAM-1 (intercellular adhesion molecule 1) is highly expressed on APCs and serves as a ligand to LFA-1[36] with extracellular domains protruding ~20 nm from the APC surface[37]. Shortly after TCR ligation, the cytosol protein talin binds to the CD18 cytoplasmic tail to convert the bent LFA-1 form to the closed headpiece conformation (inside-out signaling)[38]. ICAM-1 has a low affinity to the closed headpiece form of LFA-1 and binding promotes its opening (mediating outside-in signaling), allowing talin, kindling, and Rap-1-RIAM complexes to bind the cytoplasmic domains of LFA-1 and change its conformation to the open headpiece form that has a much higher affinity to ICAM-1[39]. Activated ICAM-1/LFA-1 interactions span an intercellular distance between the APC/T cell interface of ~36–45 nm[40,41] (Fig. 1a).

Many proteins possessing large glycosylated extracellular domains, the glycocalyx, insulate T cells from other cells and can be thought of collectively as repellers[40,42–44]. CD43 has a long extracellular domain that extends ~45 nm above the T cell surface[45] and has important roles in T cell activation and migration (reviewed in ref. [42]). CD44 is another protein with a large extracellular domain that stabilizes T cell interactions with endothelial[46] and dendritic cells (DCs)[47]. Two protein tyrosine phosphatases (PTPs) CD45 and CD148, act as repellers due to their large extracellular domains but have a special role in TCR triggering due to their catalytic activity. CD45 possesses two cytoplasmic PTP domains, and an extracellular domain composed of a mucin-like N-terminal region, a cysteine-rich domain (D1), and fibronectin type 3 (FN3) domains (D2–D4)[48]. Alternative splicing of three variable exons, A–C, encoding additional mucin-like segments to extend the N-terminal mucin domain, leads to expression of multiple isoforms: $CD45R_O$ with no variable exons and a set of $CD45R_A$, $R_B$, $R_{AB}$, $R_{BC}$, and $R_{ABC}$[49,50]. Naïve and activated human T cells express mostly $CD45R_B$ and $CD45R_O$, respectively, whereas $CD45R_{ABC}$ (B220) is highly expressed on B cells. D1–D4 domains of $CD45R_O$ are rigid and span ~21.6 nm measured by electron microscopy[48]. Variable angle total internal reflection fluorescence (TIRF) microscopy and cell surface optical profilometry measurements are consistent with an average height of ~25 nm for $CD45_{ABC}$[51]. However, the average position of the N-terminus may not accurately reflect the effective bulkiness of the highly flexible mucin-like domain. For our model, we assume a height of 25 nm for $CD45R_O$ and 35 nm for $CD45R_B$ based on the maximum extension at 0.2 nm per residue for the mucin-like N-terminus (Fig. 1a). The first critical role CD45 plays in T cell activation is to maintain Lck activity at the plasma membrane by counteracting the inhibitory kinase CSK (C-terminal Src Kinase)[52–54]. CD148 has an estimated extracellular domain length of ~47–55 nm[55]. CD148 cannot replace the function of CD45 in activation of Lck, but both phosphatases help maintain the low level of TCR complex phosphorylation in quiescent T cells. Interestingly, glycoproteins along with other components of the glycocalyx can bend the plasma membrane to form structures that facilitate interaction with the local environment for vesicle formation and release[56].

TCR triggering is initiated in small points of contact between T cells and APCs or activating surfaces[57,58]. Projections like filopodia and lamella have significant active curvature due to the underlying F-actin skeleton[59,60]. The initial triggering leads to the rapid expansion of the interface and the formation of an immune synapse[61,62]. The spreading process collapses F-actin protrusions that initiated TCR signaling, but distinct F-actin linkages to TCR microclusters through ERM proteins and LFA-1 through talin mediate transport toward the synapse center[59,63–68]. The mature immune synapse divides the interface into three distinct zones: an innermost central supramolecular activation cluster (cSMAC) that receives TCR microclusters and also includes a secretory synaptic cleft, the peripheral SMAC (pSMAC) region that is enriched in LFA-1 and traversed by TCR microclusters, and the distal SMAC (dSMAC) region which contains repellers[64], and a *corolla* of close CD2-CD58 interaction domains that can recruit multiple costimulatory receptor-ligand interactions[69] (Fig. 1b). The major opportunities to create passive membrane bending effects are at the interfaces between TCR and LFA-1 microclusters and between the CD2 corolla elements and the repellers, including CD45, in the dSMAC (Fig. 1b, c).

**Mechanical force can initiate signaling through the TCR.** The first direct evidence that mechanical forces can trigger TCR signaling was provided more than a decade ago. Kim and colleagues used optical tweezers to move beads coated with an anti-CD3 antibody for mechanical coupling with T cells. This force, applied to the TCRs through non-stimulatory antibodies attached to paramagnetic beads, causes robust calcium flux and drives ERK and MAPK phosphorylation, suggesting that the TCR complex is a mechanosensor[16]. Around the same time, Roffler's lab showed that T cells immobilized on artificial APCs that express elongated anti-CD3 antibodies are not activated unless external shear or pulling forces are applied to the T cells, suggesting that mechanical forces need to be transmitted through the TCR for effective T cell activation[18] (Fig. 2a–i). Pushing or pulling TCRs by pMHC or anti-CD3 antibodies coated on an atomic force microscopy cantilever also causes robust $Ca^{2+}$ flux in T cells[70]. The potency of tangential (shear) forces as compared to normal (pulling) forces was also described in detail by Feng et al.[71]. These results might be explained by geometry, in which vertical tensile forces are equally distributed over ligated TCRs, which requires higher force input to induce TCR conformational changes and initiate triggering. By contrast, shear forces may pull TCRs or TCR nanoclusters in one direction. Therefore, less force input is required over fewer TCRs to induce triggering. In addition, vertical forces may pull TCRs away from the cytoskeleton beneath the plasma membrane to a greater extent than shear forces, which may negatively impact triggering during vertical pulling since the cytoskeleton acts as a scaffold for signaling complexes[71].

Mechanical forces acting on TCRs may also increase antigen discrimination. In a study measuring single-bond lifetimes between TCRs and pMHC complexes during application of constant force to dissociate bonding, agonist peptides display prolonged bonding lifetimes to TCRs and maximal $Ca^{2+}$ flux under mechanical load as compared to non-loaded conditions or with antagonist peptides that lacked the ability to sustain bonding under force[72]. Similar results were obtained in a study using a micropipette to capture single red blood cells coated with weak agonists or antagonists. Under defined force, agonists act as "catch" bonds with TCRs to resist rupture whereas antagonist peptides act as "slip" bonds that are easily ruptured by force before T cells can be activated[73]. Similarly, weak agonist pMHCs and anti-CD3ε scFv connected to a DNA probe only transiently ligated TCRs and failed to activate OT-I T cells while strong agonist peptides produced stable bonds, increased tension, and TCR triggering[74]. A tension map of individual pMHC/TCR complex binding showed that seconds after ligation, T cells re-modulate their actin cytoskeleton to transmit 12–19 pN of internally generated force to engaged TCRs[75]. ICAM-1/LFA-1 binding intensifies the forces around ligated TCRs. This

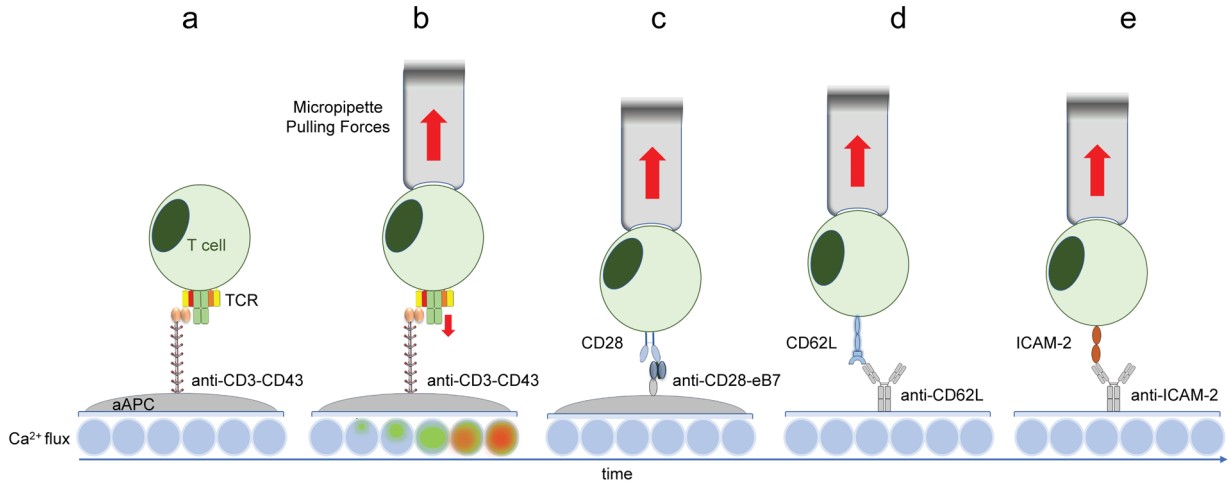

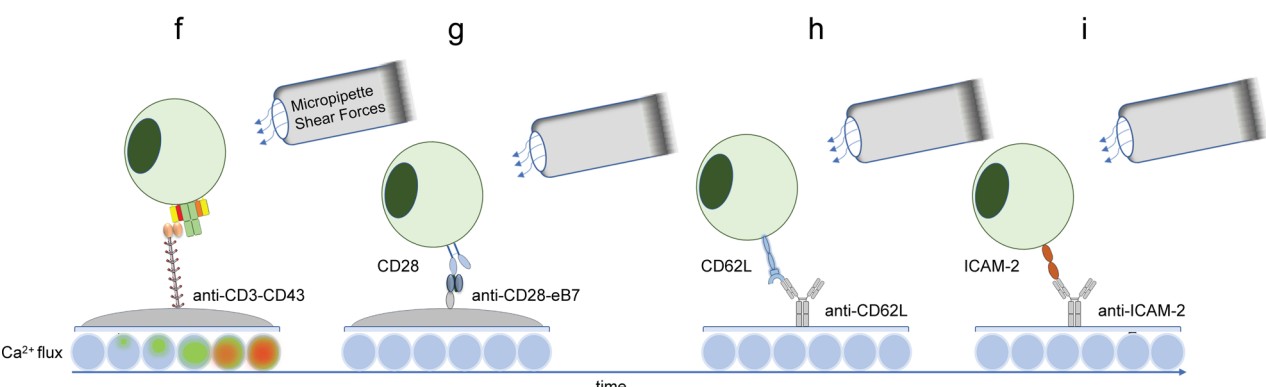

**Fig. 2 TCR triggering by mechanical forces. a** 3T3 cells were engineered to express surface-bound anti-CD3 scFv elongated with a CD43 tether (anti-CD3-CD43) that allows T cell engagement without inducing TCR triggering (illustrated by a heat map of $Ca^{2+}$ flux). **b** Applying force to T cells bound by anti-CD3-CD43 with a micropipette triggered TCR signaling as measured by $Ca^{2+}$ flux. **c–e** Similar pulling forces applied to T cells engaged via CD28, CD62L or ICAM-2 did not induce Ca flux. **f** Shear forces applied to T cells engaged via CD3 induced $Ca^{2+}$ flux. **g–i** T cells engaged via CD28, CD62L or ICAM-2 did not induce T cell activation.

mechanism is suggested to set a force threshold of approximately 12 pN for productive TCR triggering, which may act as a mechanical checkpoint to discriminate between agonist and antagonist peptides[75].

Accumulating evidence indicates that mechanical forces acting on engaged TCRs cause alterations in the relative positions of TCRs and the CD3 complex, resulting in conformational changes that favor greater accessibility of Src kinases to phosphorylate CD3 ITAMs. Each α and β chain of a TCR consists of extracellular membrane-distal variable ($V_\alpha$ and $V_\beta$) and membrane-proximal constant ($C_\alpha$ and $C_\beta$) regions, a transmembrane domain, and a short cytoplasmic tail (reviewed in ref. [76]). Conformational changes in the cytoplasmic domains of CD3 molecules appear to be transmitted *via* mechanically-induced changes in the relative orientation of the TCRα and β chains with respect to associated CD3 molecules. The FG loop composed of 12 amino acid residues connects the TCR $V_\beta$ and $C_\beta$ domains and allows the transmission of mechanical signals to intracellular signaling proteins. It also allosterically determines the strength/lifetime of pMHC/TCR bonds and controls peptide discrimination via force-driven conformational transitions[77,78]. Along similar lines, Brazin and colleagues demonstrated that the transmembrane domain of the TCRα-chain consists of two helices separated by a dynamic hinge[79]. The positively charged residues R251 in the first helix and K256 in the second helix facilitate the assembly of TCRαβ and CD3 complexes. Force applied to the TCR alters the TCRα

transmembrane conformation, thereby disrupting the association of R251 with ζ cytoplasmic chains, resulting in increased accessibility for ζ phosphorylation[79]. Taken together, these studies suggest that mechanical forces acting on engaged TCRs cause specific alterations in the orientations between TCRαβ and associated CD3 and ζ polypeptides. However, some of these allosteric changes may be initiated by zero-force interactions with pMHC that destabilize the complex[80]. The observation of zero-force allostery in the TCR doesn't preclude a role for pN forces in testing pMHC/TCR interactions kinetically, for which there is mounting evidence. For example, LFA-1 is well accepted to respond to pN forces, but nonetheless, high concentrations of soluble ICAM-1 (50 μM) can induce conformational changes in LFA-1 at zero-force[81].

### Sources of mechanical force acting on engaged TCRs

*Surface receptor topology.* An individual TCR extends about 7.5 nm above the T cell surface[22] (each Ig domain has a size of ~3.5 nm)[82], tilted at an angle of 60°[23]. MHC class I and II molecules also possess two extracellular Ig-like domains. The local membrane surrounding individual TCR and pMHC molecules must therefore approach within about 15 nm for productive engagement[40]. As detailed in the introduction, T cells express an abundance of membrane proteins with relatively large extracellular domains as compared to a TCR complex. The membranes

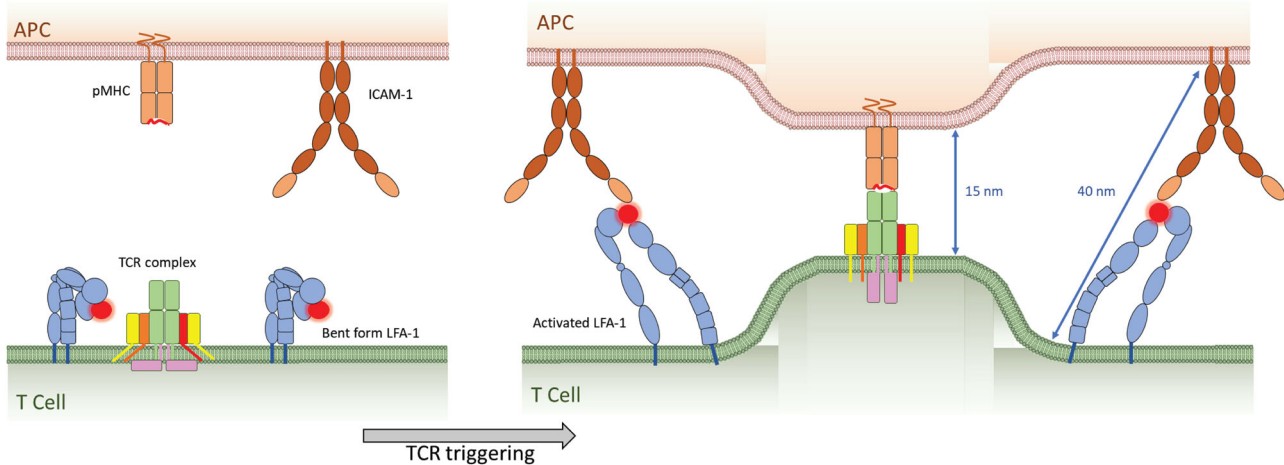

**Fig. 3 Forces exerted from ICAM-1/LFA-1 engagement.** Naïve T cells express LFA-1 in an inactive bent form. Inside out signaling induced upon TCR triggering converts LFA-1 to an open form that can ligate to ICAM-1 on APCs, generating a surface tension of 40 nm intercellular space around engaged pMHC/TCR complexes that span only 15 nm, which provide defined molecular forces around engaged TCRs leading to membrane bending.

surrounding engaged pMHC/TCR complexes must transiently curve outward away from the bulk membrane to physically accommodate nearby larger membrane receptors and adhesion molecules[83].

Biological membranes often form highly curved structures as exemplified by intracellular transport vesicles (50–100 nm in diameter), exosomes (30–150 nm in diameter), and synaptic vesicles (40 nm in diameter)[84]. Biological membranes do not readily stretch or change thickness[85] but require force to overcome an elastic resistance to bending as the inner surface is compressed and the outer surface is expanded[86,87]. Surface receptors with large extracellular domains that are not linked to the cytoskeleton diffuse away from sites of close membrane contact in response to the pressure gradient[88]. By contrast, engaged receptor pairs such as ICAM-1/LFA-1 may enforce membrane bending during TCR engagement. Immediately after TCR triggering, LFA-1 is activated through inside-out signaling and adopts an extended conformation that binds ICAM-1 on APCs at a 45° angle due to F-actin-based forces that maintain an intercellular space between the T cell and APC interface of ~36 nm[40,41,89,90]. Newly engaged pMHC/TCR complexes form in the pSMAC and are surrounded by an LFA-1 integrin ring that colocalizes with focal adhesion molecules paxillin and Pyk2 and is supported by myosin II, at least under conditions of low pMHC density[89]. Outside-in signaling by ICAM-1 engagement of LFA-1 promotes F-actin rearrangement and recruitment of LAT and SLP76[91]. Disruption of the LFA-1 ring impairs TCR microcluster generation and movement[89] and impedes TCR signaling[92]. The difference in the dimensions of ligated pMHC/TCR complexes (~15 nm) and surrounding ICAM-1/LFA-1 molecules (~36 nm) provides tensive forces that transiently bend the membrane around TCR microclusters. LFA-1 conformational changes induced by initial TCR signaling may thus act as a molecular spring to provide defined forces on the engaged TCRs and such forces might be a source for amplifying T cell activation (Fig. 3). Other surface receptors that are not freely diffusible may also contribute to membrane bending. For example, CD45 lateral mobility decreases in activated cells, in a process regulated by the spectrin–ankyrin cytoskeleton[93]. In this case, CD45, and particularly its longest forms, may also contribute to membrane bending at sites of pMHC/TCR engagement.

*Thermally-induced stochastic membrane fluctuations.* Recent advances in cell biology show that the plasma membrane of all cells continuously fluctuates in the z dimension with an amplitude of several tens of nanometers and at very high frequencies of a fraction of a second due to thermal fluctuations or stochastic membrane displacement[85,94–96] (Fig. 4a, b). T lymphocytes display continuous membrane undulations that increase after contact with an activating planar surface coated with anti-CD3 antibodies, suggesting that these movements are modulated by T cell activation[97]. Rapid nanometer-scale fluctuations of the T cell and APC plasma membranes provide sufficient membrane curvature to allow temporary pMHC/TCR engagement in the presence of dominant and larger ICAM-1/LFA-1 interactions[83]. Computational models also suggest that thermal fluctuations affect bond lifetime and discriminate low and high-affinity ligands by force-driven membrane bending[98]. Higher affinity ligands show prolonged binding, thereby generating stronger ITAM phosphorylation while low-affinity ones cannot withstand increased mechano-forces and do not lead to T cell activation[72,99]. Along the same lines, recognition of the ubiquitous marker CD47 by SIRPα ligand on the surface of macrophages requires mechanical forces generated by membrane fluctuations[100]. These studies support a model in which rapid and continuous nanometer-scale membrane fluctuations provide local membrane curvature to allow initial contact of TCRs and pMHC molecules, help T cells discriminate ligand affinity, and generate forces to trigger TCR signaling.

**Role of membrane bending in T cell activation.** The TCR complex cytoplasmic tails are associated with or buried inside the inner leaflet of the plasma membrane in resting T cells. This was first shown experimentally in lipid vesicles, where the cytoplasmic domain of ζ was found to associate with acidic lipids, thereby hindering access of ζ ITAMs to Src phosphorylation[101]. Förster resonance energy transfer experiments in live cells further demonstrate that basic residues present in the cytoplasmic tail of CD3ε are in close association with acidic phospholipids present in the inner leaflet of the plasma membrane, resulting in the embedding of CD3ε ITAMs inside the plasma membrane where they are inaccessible to Lck phosphorylation[102]. Amino acid substitutions that decrease the association of CD3ε with the plasma membrane significantly decrease TCR surface expression, impair localization of the TCR complex in the immune synapse and reduce overall TCR signaling in transgenic mice[103]. Mutating basic amino acids in ζ chains also impairs their association with

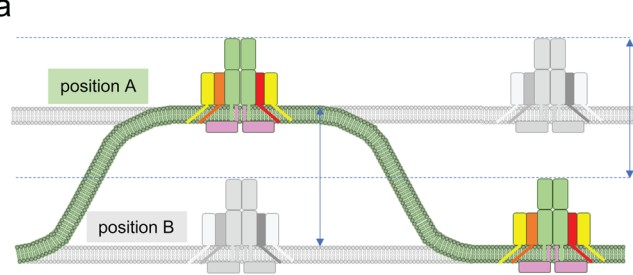

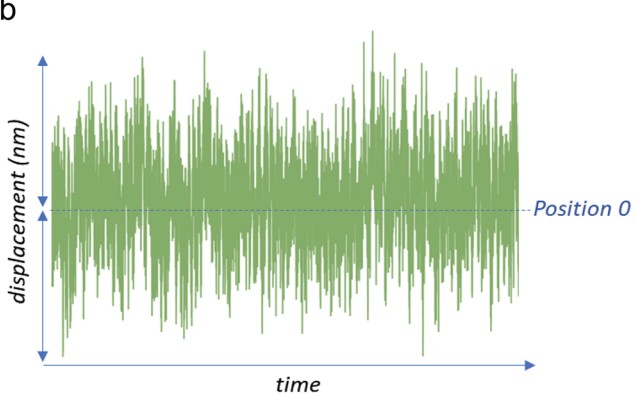

**Fig. 4 Stochastic membrane displacement. a** The plasma membrane of T cells (and all cells in the body) continuously fluctuates to dissipate thermal energy at physiological temperatures. TCR receptors, therefore, are normally moving up and down on the T cell surface (colored and gray positions A and B). Engagement of TCRs by pMHC on an APC fixes the relative positions of the TCR and pMHC, which opposes the normal membrane fluctuations and provides another source of mechanical force on engaged TCRs. **b** illustrated displacement of position 0 on the nm scale over a unit of time.

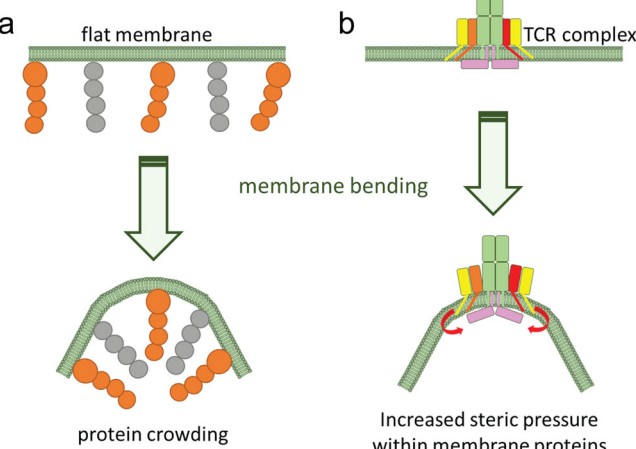

**Fig. 5 Protein crowding. a** Proteins associated with biological membranes occupy large surface areas on flat membranes (left), but are crowded in the inner leaflet when the membrane is bent inward (right). This creates steric pressure that opposes membrane bending. **b** Bending the plasma membrane directly around engaged TCRs may increase steric pressure on the CD3 complex and assist dissociation of buried CD3 ITAMs to the cytosol.

the plasma membrane, hinders lateral distribution, and compartmentalization of the TCR complex, and severely attenuates signaling[104].

As a biological membrane curves inward, steric pressure increases as the surface area that is available to each protein on the inner leaflet of the lipid bilayer decreases[84,105] (Fig. 5a, b). Nanometer-scale bending of the local membrane around engaged TCRs may generate sufficient steric and lateral pressure to eject TCR complex cytoplasmic domains from the inner leaflet due to protein crowding[106], this may result in TCR triggering due to phosphorylation of exposed CD3 ITAMs by constitutively active Lck molecules[101]. Membrane curvature may also dynamically alter the interactions of TCR complex cytoplasmic domains with the plasma membrane by changing lipid packing and altering local lipid hydrophobicity[107–109].

In situ proximity assays reveal that pMHC/TCR engagement induces re-arrangement of the cytosolic juxtamembrane positions of the ζζ chains. The ζζ chains are divaricated in resting T cells but binding of a pMHC to a TCR drives intra-complex apposition of these domains[17]. The mechanical transition of ζζ chains from *cis* (divaricated position) to *trans* (intra-complex apposition) is controlled by pMHC affinity; weak agonists are associated with more *cis* conformation while strong agonists drive the *trans* conformation[17]. Membrane association of the cytoplasmic tail of CD3ε also decreases in the presence of $Ca^{2+}$ due to competition for negatively charged phospholipids, suggesting a possible feed-forward loop to further sustain and amplify TCR signaling[110]. Similarly, the reduced hydrophobicity and net negative charge of phosphorylated ITAM tyrosines

negatively regulate the association of TCR complex cytoplasmic domains to the plasma membrane inner leaflet (reviewed in ref. [111]). An important role for membrane curvature in T cell activation is consistent with studies showing that plasma membrane lipids regulate T cell signaling (reviewed in ref. [112]). Signaling due to mechanical curvature of the plasma membrane may be amplified in TCR nanoclusters, which are composed of 7–30 TCR complexes in non-activated T cells[113,114], due to increased steric pressure upon TCR engagement. Indeed, tyrosine phosphorylation is rapidly propagated within nanoclusters[114]. On the other hand, there is controversy as to whether the appearance of TCR clusters in super-resolution studies is significantly different from a random arrangement of receptors[115]. Jung et al suggest that TCR nanoclusters are associated with tips of microvilli, supporting the argument that small clusters are not random[116]. It should be noted that these configurations, including pre-segregation of TCR from CD45[116], are based on fixed snapshots and may be subject to rearrangements on a subsecond time scale to prevent spontaneous signaling while keeping the system poised to respond rapidly.

One of the biggest challenges is to prove that membrane bending at the nanometer scale can be achieved by the molecular topology of the T cell membrane. In an elegant study performed by Kim and colleagues, beads coupled to liposomes or cells by biotin-streptavidin or cadherin interactions were pulled by optical tweezers to generate membrane curvature, in which the liposome radius, surface area of contact, temperature, applied forces, and membrane displacement were measured. Kim estimated that applying 9 pN of force on vesicles with a 70 nm radius in a deformation-free configuration and with a 58 nm radius of the area of contact with the underlying surface caused the membrane to extend up to ~60 nm[117]. Although a liposome is obviously different from a cell, the intercellular distance between engaged APCs and T cells needs to be less than 15 nm, suggesting that 9 pN of force may be sufficient to induce membrane bending at engaged TCRs. In another study, Liu applied forces to individual TCRs to generate a tension map showing that about 12 pN of force acting on individual TCRs can initiate TCR signaling[75]. Taken together, these studies indicate that the amount of force needed to generate membrane deformation is similar to the force

required for TCR triggering. Hatzakis et al. developed an interesting method to measure curvature in deformed membranes based on the insertion of wedges of alkyl motifs of membrane curved proteins such as the N-terminus of endophilin A1 (eAH); A higher degree of membrane deformation creates more binding sites for the wedges[118]. Interestingly, endophilin A2, a closely related endophilin protein that regulates membrane bending through a BAR domain[119], colocalizes with TCR microclusters and pZap70 upon TCR triggering. This protein also plays an important role in TCR internalization, and its deficiency reduces T cell activation significantly[120].

## Mechanical membrane deformation and TCR/CD3 reorganization explain experimental observations of T cell activation

*Soluble monovalent ligands do not activate T cells.* Early experiments on T cell activation revealed that soluble monovalent TCR ligands are unable to activate mouse T cell hybridomas. However, when the same ligands are tethered to surfaces, strong T cell activation is achieved[121–125]. For example, Wu and colleagues showed that anti-CD3ε, anti-TCRα, or anti-TCRβ Fab do not activate T cell hybridomas. By contrast, covalent attachment of these Fab to an anti-MHC-II Fab allowed T cell activation in the presence of cells that expressed MHC-II, indicating that T cell activation by monovalent ligands occurs if they are fixed on a surface[121]. Similarly, soluble monovalent MHC molecules do not activate T cells[126], whereas MHC molecules chemically coupled to iron–dextran nanoparticles effectively activate T cells[122,123]. Soluble anti-CD3ε antibodies activate T cells if coupled to other cells such as DCs or monocytes via Fc receptors[124,125]. In fact, anti-CD3ε antibody-coated beads are now commonly used for efficient T cell activation and expansion[127]. Therefore, successful T cell activation is achieved if pMHC or anti-CD3ε antibodies are coated on surfaces, artificial APCs, SLB, or beads[99,128–135] (Fig. 6a). A caveat is that biochemical methods with high-affinity pMHC demonstrate allosteric changes in TCRs with monovalent engagement by soluble ligands[80]. The consistent observation of robust signaling by surface presented pMHC or anti-CD3 can be explained by deformation of the T cell membrane to allow a close approach of TCRs to immobilized ligands, which is not required with soluble ligands. Likewise, membrane fluctuations around a TCR bound to an immobilized ligand generate substantial mechanical forces in comparison to a soluble ligand that can move with the TCR membrane. These results are consistent with both the KS model and with the deformation of the T cell membrane by physical forces. In fact, the KS model is very popular because of the obvious segregation of CD45 from the immune synapse and TCR microclusters, and studies showing that disruption of CD45 segregation abolished signaling[133,134,136]. However, other studies show that CD45 segregation is not required for TCR triggering[135,137,138].

*Constrained dimeric ligands trigger TCR signaling.* In contrast to monovalent ligands, multivalent ligands, formed by secondary antibody cross-linking or by the addition of synthetic linkers, can activate T cells[139–141]. For example, soluble monovalent MHC-II molecules do not induce T cell signaling, unless they are first physically linked as dimers[126]. This observation has been explained as showing that dimerization is sufficient to initiate signaling, in analogy to many receptors that transmit signals upon dimerization[142]. In addition, it was believed that TCR clustering is important for TCR triggering by inducing changes in the CD3 cytoplasmic tails and that TCR clustering is a prerequisite for their activation[143,144]. However, monomeric pMHC and TCR molecules on APC and T cells are capable of triggering signaling[145]. In addition, TCRs are present on naïve T cells and microvilli as pre-formed TCR microclusters and proximal

signaling proteins (Lck and LAT) for fast-act sensing/activation responses[67,113,139,146–149]. An alternative explanation is that divalent and oligomeric ligands trigger T cell activation if they change the orientation of the TCR relative to CD3 or the local plasma membrane. This implies that an effective bivalent ligand must be rigid and span a distance shorter than the natural distance between two TCR complexes. Indeed, strong T cell activation is achieved by soluble pMHC dimers linked *via* a short spacer, while extending the linker length between pMHC dimers abrogates TCR triggering[150] (Fig. 6b).

*TCR triggering requires membrane-bound ligands with particular dimensions.* Several labs demonstrated that the dimensions of membrane-tethered ligands affect TCR triggering. Anti-CD3 scFv anchored on fibroblasts via tethers composed of 1 or 2 Ig-like domains effectively activate T cells whereas anchoring the same anti-CD3 antibody via a longer tether derived from the extracellular domain of CD44 completely prevents T cell activation, even though TCR binding is not compromised[151]. Elongation of pMHCs tethered on the surface of artificial APCs progressively reduces their ability to activate T cells to secrete IL-2[99,133,134]. Elongating pMHC with polyethylene glycol tethers also reduces their potency to activate T cells[152]. Along similar lines, chimeric antigen receptor T (CAR-T) cells engineered to bind a distal epitope of CD22 less efficiently kill cancer cells compared to CAR-T cells targeting a CD22 proximal epitope[153]. A low affinity ($K_d$ ~10 μM) bispecific T cell engager (biTE) antibody that binds to CD3 and chondroitin sulfate proteoglycans on melanoma cells also more effectively activates cytotoxic T cells to kill melanoma cells when bound to a membrane-proximal epitope as compared to a membrane-distal epitope[154]. These results are consistent with the greater mechanical force generated by small ligands versus elongated ligands due to greater curvature of the plasma membrane to allow engagement of TCRs.

The dimensions of the intercellular space between T cells and APCs that leads to the best activation is equivalent to ~4 Ig-like domains (~15 nm)[99]. At this distance, the affinity of agonist pMHC appears to be sufficient to overcome the repulsive forces of the glycocalyx on the surface of opposing T cells and APCs and generate sufficient forces to trigger TCR signaling. This distance is naturally generated when pMHC, with an extracellular region composed of 2 Ig-like domains ligates to TCRαβ that also has an extracellular region composed of 2 Ig-like domains. Thus, artificial APCs that express membrane-tethered anti-CD3 scFv with an extracellular region corresponding to 3 Ig-like domains generate optimal T cell activation when bound to CD3ε, which contains an extracellular region composed of a single Ig-like domain. Anti-CD3 scFv anchored to the surface of APCs with tethers longer than 4 Ig-like domains produce less T cell activation, and signaling is completely absent for tethers of 6 or more Ig-like domains[99] (Fig. 7a). Moreover, reducing the APC/T cell intercellular space negatively impacts TCR triggering, likely due to the need to overcome an increasingly steep energy barrier for productive binding as the glycocalyx is compressed at the interface between T cells and APCs[99]. In fact, decreasing the tether dimensions to zero Ig-like domains by directly anchoring the scFv on the APC surface results in loss of T cell binding, presumably due to the steric hindrance of the glycocalyx[151].

We posit that this defined intercellular distance exerts sufficient mechanical forces on engaged TCRs due to membrane bending to achieve TCR triggering. Releasing these repulsive forces by ligand elongation results in loss of T cell activation because TCR engagement can occur without membrane bending and generation of mechanical forces.

*Elongated high-affinity ligands can activate T cells.* In contrast to pMHC and low-affinity anti-CD3 scFv, elongation of the tether

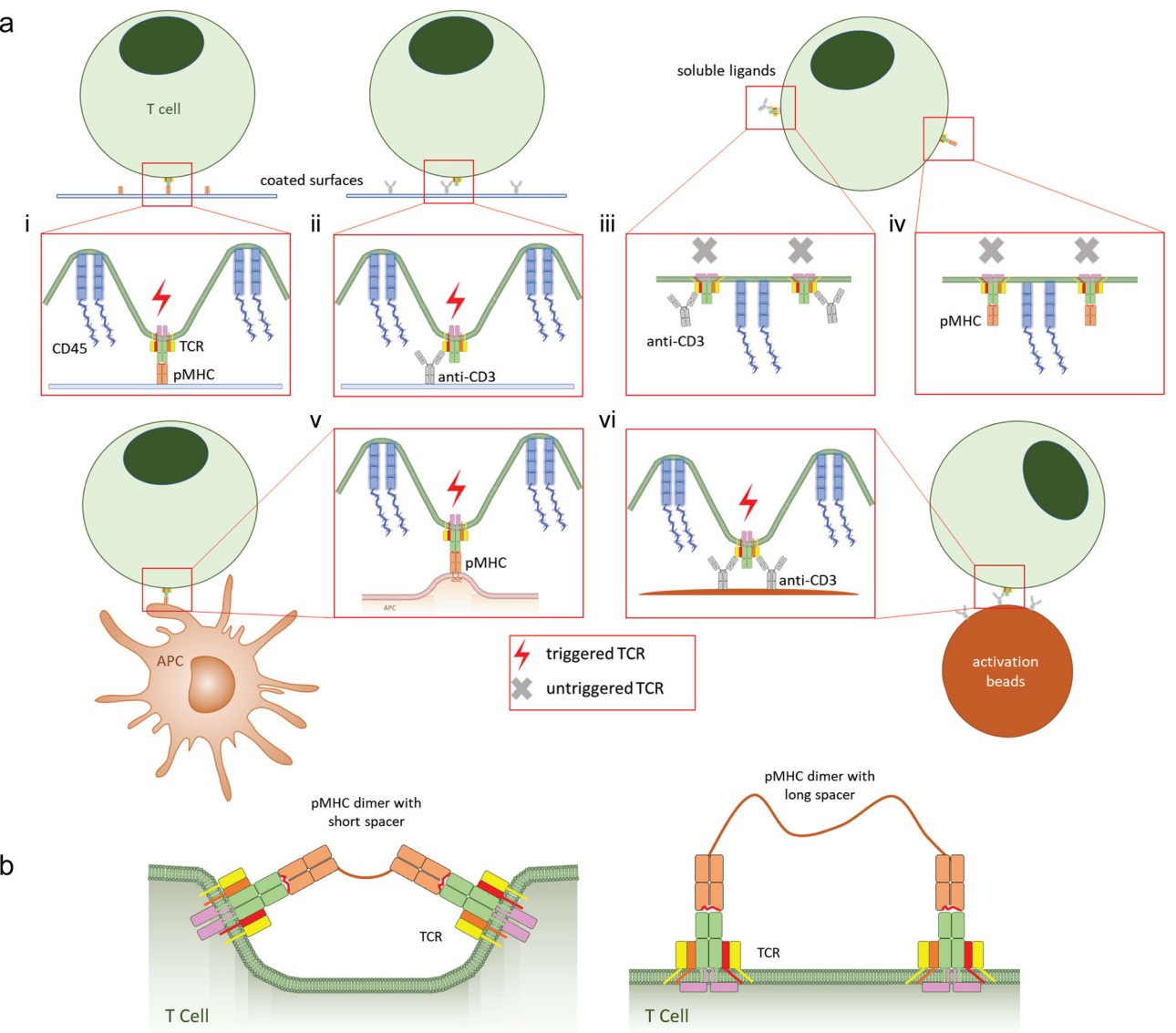

**Fig. 6 T cells can be activated by tethered ligands. a** T cells are activated by ligands coated on surfaces such as glass or plastic, activation beads, or APCs. Ligand binding bends the membrane around TCRs to physically accommodate membrane proteins with large ectoplasmic domains. Soluble ligands are poor T cell activators because they bind TCRs without inducing plasma membrane bending: **i** pMHC-coated glass surface. **ii** anti-CD3 antibody-coated glass surface. **iii** soluble anti-CD3 antibodies. **iv** soluble pMHC. **v** APC expressing pMHC. **vi** activation beads coated with anti-CD3 antibodies. **b** TCR triggering by soluble pMHC can be achieved with pMHC dimers linked via short spacers (left) but not when the pMHC dimers are linked via a long spacer (right). Only short spacers are able to induce membrane bending.

does not adversely affect activation of primary human T cells for membrane-anchored high affinity OKT3 anti-CD3 scFv[99]. Independence from tether dimensions depends on antibody affinity because mutations that reduce OKT3 scFv affinity restore the dependency of T cell activation on tether dimensions[99]. Similar results are observed when ligands are tethered to an SLB surface; a low-affinity variant of OKT3 anchored to the SLB via an elongated tether is unable to trigger calcium flux, Zap70 phosphorylation, T cell proliferation, or IL-2 secretion whereas the same scFv tethered via one Ig-like domain produces strong T cell activation[135] (Fig. 7b).

Ligands with elongated tethers generate less mechanical forces due to reduced topological differences in the sizes of engaged TCRs and surrounding large membrane proteins. Thermal membrane fluctuations may therefore represent the predominant sources of mechanical force for elongated membrane-anchored TCR ligands. High-affinity ligands can bind to TCRs for extended periods without rupture, so even when elongated, they can accumulate sufficient thermally-induced mechanical force on engaged TCRs to achieve T cell activation. By contrast, low-affinity elongated ligands bind for insufficient times to trigger signaling, consistent with a model proposed by Pullen and Abel[98]. Loss of entropy upon binding of elongated low-affinity ligands also effectively reduces their binding affinity, contributing to poor agonist activity[155].

*TCR triggering is primed in the d/pSMAC and terminated in the cSMAC.* Early studies proposed that TCR microclusters that accumulate at the cSMAC sustain TCR triggering in the mature synapse[156,157]. However, this concept was revised when the dynamics of immunological synapse formation were further dissected using TIRF imaging of live cells. TCR microclusters initially contact agonist pMHC in the pSMAC and then start to move radially to the center of the APC-T cell contact[158] while

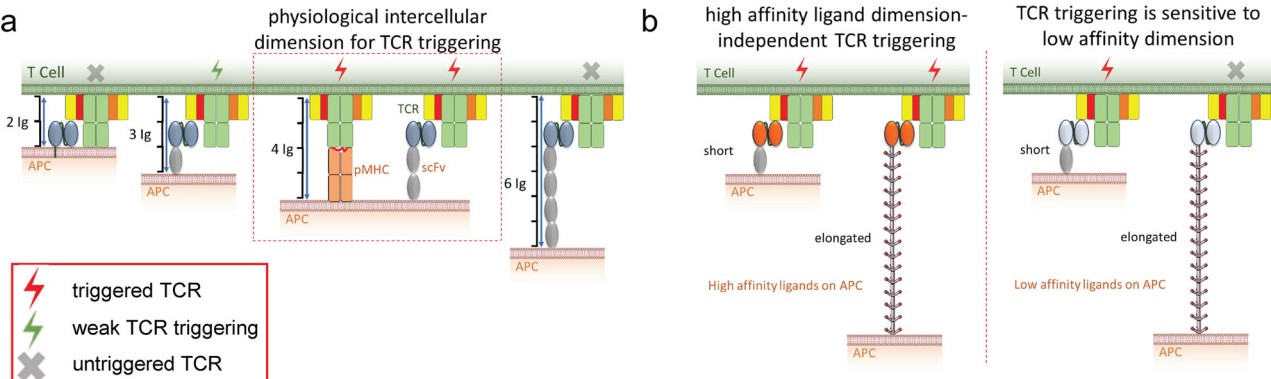

**Fig. 7 T cell activation requires ligands with particular dimensions. a** TCR triggering is achieved when the interspatial distance between T cells and APCs is around physiological dimensions which is 4 Ig-like domains (~15 nm). Similar results are obtained using artificial APCs expressing membrane-bound anti-CD3 scFv elongated with different tethers. **b** High-affinity TCR ligands tethered on artificial APCs can trigger TCR signaling independent of their dimensions (left) while low-affinity TCR ligands lose their potency to activate TCR when elongated.

new activated TCR microclusters continually form in the dSMAC and then translocate through the pSMAC[159]. ζζ and Zap70 are first phosphorylated in TCR microclusters in the pSMAC and are sustained for minutes as they translocate through the pSMAC, indicating that TCR signaling is initiated in protrusion at the out-edge of the immunological synapse, whereas TCR signaling is terminated in the cSMAC[160]. This phenomenon is particularly important for CD4 helper T cells that deliver CD40L to APCs through the release of extracellular vesicles that also contain TCRs, thus terminating signaling[158].

TCRs ligation in the pSMAC can generate strong mechanical forces due to differences in the dimensions of engaged TCRs and large extracellular proteins that are highly enriched in this region[161]. Productive TCR signaling is dependent on the formation of a micro-adhesion ring of LFA-1 bound to ICAM-1 surrounding the TCR microclusters[89]. These microsynapses are continuously generated in the pSMAC and move to the center of the immune synapse where signaling is terminated[89,162]. The defined dimensions of activated ICAM-1/LFA-1 on T cells and APCs, respectively, may fix the intercellular distance between these cells to generate the optimal level of mechanical force for agonist peptide MHC complexes[35,163]. By contrast, the cSMAC may produce less mechanical force on engaged TCRs because many small adhesion molecules with similar dimensions are highly enriched in this region; this may also dampen the effects of stochastic membrane fluctuations on nearby engaged pMHC/TCR clusters as little topological forces are generated. Furthermore, the endosomal sorting complexes required for transport appear to actively terminate TCR signaling in a manner dependent upon ubiquitination.

The importance of membrane bending induced by steep gradients in the molecular dimension of receptors and repeller is supported by the CD2 corolla in the dSMAC[69], which is positioned at the origin of TCR microcluster formation. CD2/CD58 corollas surrounding TCR microclusters in the dSMAC are associated with enhanced phosphorylation of Src family kinases, LAT and PLC-γ. CD2–CD58 interactions in the cSMAC are less efficient at amplifying TCR signals[69]. Interestingly, CD2–CD48 interactions actually decrease the 2D affinity for pMHC/TCR interactions, although they still promote pMHC/TCR interactions at physiological pMHC densities[164]. CD2–CD48 interactions span 3 Ig-like domains at the intercellular space between T cells and APCs due to face-to-face interaction of the N-terminal Ig domains of CD2 and CD58/CD48[34]. Increasing the length of CD48 by 2 or 3 Ig-like domains decreases the efficiency of adhesion by 10-fold on SLB. In addition, elongating CD48 on

APCs disrupts ICAM-1 mediated costimulation of pMHC recognition[155]. We speculate that natural CD2 interactions to CD48 or CD58 amplify pMHC/TCR mediated membrane bending by working against glycocalyx repulsion and this effect is further intensified by ICAM-1/LFA-1 cross-bridges in the pSMAC. The data are further consistent with the interpretation that elongated forms of CD48 lack sufficient binding energy to impact membrane bending mediated by pMHC/TCR interactions, but can dissipate membrane bending between sites of pMHC/TCR and ICAM-1/LFA-1-interactions to eliminate costimulatory effects.

*Close contact can trigger TCR signaling.* T cells can be activated by forced spreading on a surface coated with non-binding immunoglobulins as shown by calcium mobilization and expression of very early markers of TCR triggering[48]. This finding was explained based on the kinetic segregation (KS) model[165], in which spatially-mediated segregation of the large CD45 membrane receptor phosphatase from pMHC/TCR at sites of close contact between T cells and the solid surface allows prolonged phosphorylation of CD3 ITAMs[48]. The absence of CD45 at the close-contact sites where signaling started was demonstrated by TIRF microscopy. Jurkat and primary T cells that spread over poly-L-lysine also display activation as measured by calcium mobilization, Zap70 phosphorylation, and CD69 upregulation to comparative levels as OKT3 anti-CD3 antibody-induced activation[166].

Forcing cells to spread in close contact with a solid surface can mechanically compress membrane receptors and adhesion molecules on the surface of T cells. This physical compression forces lateral movement of large surface proteins away from sites of close contact between the T cell membrane and the solid surface, as observed in the above studies[48,166] as well as in studies of protein sorting due to size differences on cell membranes[167]. Physical compression of large surface proteins and receptors also creates an inward force on the T cell membrane, resulting in membrane curvature relative to short receptors, such as TCRs, that do not contact the surface.

*CD45 segregation is not required for TCR signaling.* Many studies have tested the KS model by elongating pMHC or truncating the CD45 extracellular domain[99,134,168–170]. However, the interpretation of these studies is complicated because CD45 both positively and negatively regulates the activity of Lck through its two tyrosine residues (Y394 and Y505). CD45 dephosphorylates Y394 and converts fully active Lck to a basally active form, which

can be blocked by the Csk phosphorylation of Y505. CD45 can also dephosphorylate Y505 and convert Lck to the basally active state again[49]. Therefore, any modifications that alter the localization of CD45 relative to Lck and TCRs may affect Lck activity and T cell activation in a complicated fashion. Besides, the close contact between T cells and APCs that provides the driving force for lateral diffusion of large CD45 molecules away from engaged pMHC/TCR complexes also generates mechanical forces around the engaged TCRs[171].

The difficulty in interpreting CD45 segregation studies is illustrated by the finding that truncation of the extracellular domain of CD45 reduces TCR triggering by both immobilized and soluble anti-CD3 antibodies[170]. Since soluble antibodies cannot provide a compressive force to drive segregation of CD45 away from areas of close contact between T cells and APCs, these results may reflect the association of the CD45 extracellular domain with glycosphingolipid-enriched membranes (GEMs) where Lck is concentrated, whereas truncated CD45 chimeras are less associated to GEMs[170]. Other studies have replaced the extracellular domain of CD45 with ligands such as CD2 or CD86, which can bind to receptors on APCs[54,168]. However, binding or cross-linking of CD45 can reduce phosphatase catalytic activity and block T cell activation[168]. Replacing the large CD45 ectodomain with small extracellular domains derived from EGFR or MHC-I, which should not segregate based on size, also allows T cell activation[168,169]. These studies suggest that segregation of CD45 based purely on extracellular domain size is not required for the initial triggering of TCRs. TIRF microscopy confirms that TCRs are efficiently triggered by elongated, high-affinity ligands in the presence of colocalized CD45[135].

Studies that examine CD45 segregation by extending the dimensions TCR ligands are limited because these tethers are flexible and tilt in the interface leading to an increase in membrane spacing of only 1–2 nm per Ig-like domain[99,134,155]. Thus, these are "soft" extensions and likely also undergo significant size fluctuations. Cai et al. developed nanofabricated substrates on which anti-CD3 Fab was either presented directly on gold at the level of the SLB (2D) (Fig. 8a) or elevated 10 nm above the SLB on $SiO_2$ pillars (3D) (Fig. 8b)[138]. The 2D or 3D substrates presented anti-CD3 Fab in clusters with precise spacing between 40 and 150 nm in different regions of the substrate, and adhesion was provided with ICAM-1 on the SLB, which was essential for T cell responses. T cells interacting with 2D substrates with a full range of anti-CD3 spacing triggered T cell activation (Fig. 8a). By contrast, T cells interacting with the 3D substrates were only triggered at 40 nm spacing between anti-CD3 (Fig. 8b). Importantly, $CD45R_O$ was partly excluded from TCR microclusters on the 2D, but not 3D surfaces. Thus, precise positioning of anti-CD3 Fab demonstrated that CD45 exclusion is not required for TCR triggering. This 40 nm spacing required for activation by 3D surfaces is 10-fold larger than inter-TCR distances required for activation by soluble dimeric ligands[142]. It was speculated that the extended cytoplasmic domains of the TCR might start to overlap and cooperate at 40 nm, but this was not directly tested[142]. An alternative explanation is that adequate membrane curvature is only generated on 3D surfaces when the anti-CD3 Fab is less than 40 nm apart, which would also require intercalation of ICAM-1/LFA-1 interactions or similar larger complexes to force bending (Fig. 8a, b). This interpretation suggests that the degree of membrane curvature is a quantitative parameter contributing to T cell signaling similar to ligand density or the degree of CD45 exclusion.

In a reconstitution system using SLBs in which Lck was added along with proteins necessary for the signaling cascade, a functionally-active CD45 phosphatase domain was totally segregated from LAT signalosomes[137]. This was due to exclusion of negatively-charged CD45 phosphatase domains (isoelectric point 6.4) from similarly charged LAT condensates, leading to repulsion[137]. In contrast to the size-dependent exclusion of the CD45 extracellular domain, this study suggests that segregation can occur after successful TCR triggering upon establishing phase-separated domains formed by LAT and adapter proteins. CD45 may play a role in antigen discrimination. Weiss and colleagues used titrated amounts of CD45 and Csk to investigate the balance between Lck activation and ζ chain dephosphorylation, and showed that CD45 acts as a signaling gatekeeper that allows TCR triggering with the basal amount of active Lck for high-affinity peptides, whereas low-affinity peptides fail to induce signaling[172].

*Membrane mechanosensing and mechanical stiffness potentiate TCR triggering.* T cells overcome repulsive barriers including the APC glycocalyx and establish close proximity by interacting with adhesion molecules such as ICAM-1. This interactive binding initiates actin polymerization to generate T cell membrane protrusions which facilitate scanning of cognate pMHC on the APC surface[173]. These micron-scale protrusions, manifesting as "microvilli" initially or "invadosome-like protrusions" (ILPs) at later stages, allow rapid and sensitive discrimination of agonist pHMC[60,173]. ILPs have an average size of ~430 nm in-depth and ~350 nm in width that helps overcome the glycocalyx barrier and form close intercellular contacts with APCs to enhance antigen probing efficiency[173]. These protrusions are stabilized by calcium mobilization, which may support signal amplification[173].

Variations in membrane tension such as membrane stretching applied to focal adhesion or along actin fibers open mechanosensing channels (MS) to mediate intracellular $Ca^{2+}$ rise[174,175]. For instance, TRPV2 opens and mediates $Ca^{2+}$ entry in T cells when subjected to mechanical stress[176]. Oscillations of free intracellular $Ca^{2+}$ concentrations have been witnessed more frequently in engaged than in lone cells by the recruitment of MS channels to the immune synapse[177]. Piezo1 is an important mechanosensing protein that senses mechanical forces applied to the membrane and converts it to an electrical response[178]. TCR signaling is attenuated in Piezo1-knockdown T cells, and since soluble antibodies are not potent T cell activators, adding Piezo1 agonist along with anti-CD3 and anti-CD28 antibodies potentiates TCR signaling without the need for antibody immobilization or crosslinking[179]. Increasing membrane tension by nanoscale lipid compression opens Piezo1 channels[180–183]. Therefore, Piezo1 may sense membrane curvature that accompanies APC/T cell engagement, and coordinate with TCR engagement to enhance TCR signaling. In this regard, the T cell membrane is condensed at sites of TCR triggering, whereas relaxing membrane condensation by adding 7-ketocholesterol impairs the formation of signaling complexes and cytoskeleton restructuring, resulting in reduced IL-2 secretion but not $Ca^{2+}$ flux and tyrosine phosphorylation[184].

It is therefore expected that increasing membrane rigidity may require more force to generate membrane bending. Cholesterol, for example, increases membrane rigidity and negatively impacts TCR triggering. Treating human T cell lines or naïve human T cells with cholesterol upon activation with anti-CD3/anti-CD28 impairs calcium signaling and T cell activation[185]. Accumulation of cholesterol in aged mouse T cells also correlates with decreased activation potential[186,187]. Similarly, high cholesterol levels were detected in aged human T cells. At high cholesterol levels, membrane fluidity decreases leading to impaired lipid raft formation and reduced T cell activation and proliferation[188]. However, the allosteric model of TCR regulation posits that cholesterol has an affinity to TCRβ and can bind only with non-triggered TCRs, inhibiting overall TCR stimulation and that cholesterol removal from around TCRs is required to initiate

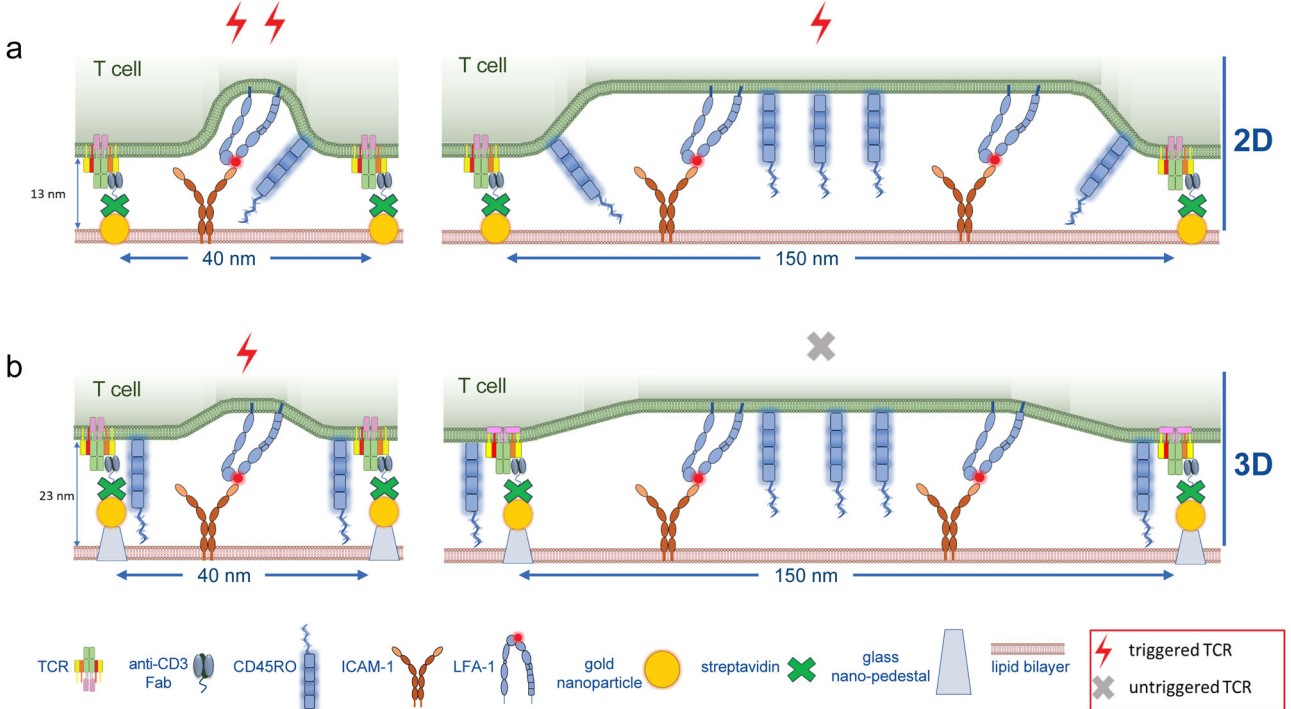

**Fig. 8 Membrane curvature and CD45 exclusion contribute to TCR triggering.** Micropatterned surfaces were prepared in 2D with gold nanoparticles (NPs) at the same level as an SLB (**a**) or in 3D with gold NPs elevated 10 nm on SiO₂ pillars above the SLB (**b**). Anti-CD3 Fab was linked to NPs and ICAM-l molecules were present on the SLB for adhesion. Large membrane curvature is generated at engaged TCR clusters in the 2D system to accommodate large ICAM-1/LFA-1 complexes (**a**) whereas elevation of anti-CD3 Fab in the 3D system generates less membrane curvature for a given ligand spacing (**b**). However, when the distance between NP pillars is reduced to 40 nm TCR signaling is restored (**b**, left). This might be explained by a high degree of membrane curvature when ICAM-1/LFA-1 interactions are interspersed with the closely spaced pMHC/TCR (**b**, left). While ICAM-1 didn't accumulate in 3D TCR clusters with 40 nm spacing, a relatively small number of interspersed ICAM-1/LFA-1 interactions would be sufficient to enforce curvature; further work is required to test for high curvature in the 3D 40 nm clusters.

signaling[189]. Similarly, treating T cells with cholesterol sulfate, a natural derivative of membrane cholesterol inhibits TCR avidity and disrupts TCR nanoclusters[190]. Yang et al. reported that inhibiting cholesterol esterification increases cholesterol levels in the membrane of CD8+ T cells and enhances their antitumor activity[191]. These diverse studies suggest that cholesterol may play a more complicated role in TCR triggering than simply adjusting membrane rigidity.

The stiffness of activating substrates affects migration, activation, gene expression, cytokine secretion, and metabolism of human T cells[192–194]. Cytokine production and CD25 expression by effector and memory human CD4 T cells increases in response to increasing stiffness of substrates bearing TCR ligands[12,195]. RNA microarray analysis of effector CD4 T cells activated for 24 h on polyacrylamide gels of varying stiffness found enhanced gene expression of different cytokines, T cell activation markers including CD69, and T-cell-specific transcription factors in response to increased substrate stiffness[12]. Expression of lamin genes[196], translation initiation factor *EIF4E*[197], and metabolism-related genes[197,198] also gradually increase with stiffness on anti-CD3-coated polyacrylamide gels. Thus, mechanical stress from stiffness-based membrane deformation acts as a rheostat, leading to differential expression of genes involved in TCR/CD3 induced T cell functions[12,199–201]. However, pMHC and ICAM-1 on SLBs over soft (4 kPa) polydimethylsiloxane supports triggered early TCR signaling similarly to SLBs on stiff (GPa) glass supports[202].

**Summary**. TCR triggering is crucial to activating T cells in immune responses and T-cell-based therapies but how pMHC/

TCR engagement initiates proximal signaling is not fully understood. Increasing evidence demonstrates that mechanical forces play a role in initiating signaling through the TCR. The TBM model posits that mechanical forces acting on engaged TCRs may originate from at least two different sources. Surface receptor topology can exert forces on small engaged pMHC/TCR pairs to accommodate nearby receptors with large extracellular domains. Thermally-induced stochastic membrane fluctuation can exert tensive forces on engaged pMHC/TCR complexes between the T cell and APC interface. In addition, cytoskeletal remodeling may transfer additional forces to tone signaling and set an energy threshold to discriminate between agonist and antagonistic peptide antigens, low and high-affinity ligands. The TBM model predicts that mechanical forces induce membrane curvature immediately around engaged pMHC/TCR complexes to overcome the energy barrier necessary to release CD3 cytoplasmic domains from the inner leaflet of the plasma membrane. Increased accessibility of the ITAMs in the CD3 cytoplasmic domains to Lck phosphorylation can then trigger the signaling cascade to activate T cells. The rapid influx of Ca²⁺ may exert a positive feedback loop by competing with negatively charged phospholipids to help release CD3 ITAMs from the inner leaflet of the plasma membrane[110].

Force may also shape the outcome of T cell activation and the magnitude of proliferation. Substrate stiffness detected by mechanosensing machinery of T cell surfaces such as Piezo1 and ion channels can regulate transcription factors and the gene expression profile associated with downstream signaling of T cells. Many of these are involved in T cell metabolism and cytokines required for effector T cell functions.

In the TBM model, we expect that TCR triggering is regulated tightly by membrane bending to reduce signaling noise. Therefore, the following are expected to happen sequentially to start TCR signaling:

1. T cells meet APCs that present an agonist peptide on MHC molecules.

2. T cell uses microvilli/ILPs to sense the availability of agonist pMHC and to overcome the glycocalyx barrier.

3. Catch bonds are formed only by agonists whereas membrane fluctuations rapidly break slip bonds to reduce signaling noise (from antagonist).

4. Stabilized catch bonds may induce cooperative binding with other TCRs.

5. The topological variations between TCRs and other membrane proteins induce membrane bending, which results in further pressure to break the non-specific binding. Conformational changes within the TCR complex components relative to the T cell membrane cause dissociation of ζ chain from the inner leaflet of the plasma membrane.

6. Initial triggering starts, phosphorylated ζ induces downstream signaling that stabilizes the structure of ILPs by calcium-dependent mechanisms. In addition, talin is recruited to induce LFA-1 activation to ligate to ICAM-1 on the APC surface.

7. ICAM-1/LFA-1 interactions induce outside-in signaling that increases LFA-1 binding affinity. ICAM-1/LFA-1 binding intensifies membrane bending to amplify TCR signaling.

8. Microadhesion rings move in arrays to the center of APC/T cell contact area forming the cSMAC, while other microadhesion rings keep appearing at the pSMAC and moving to the cSMAC powered by the cell cytoskeleton.

**Future directions**. The TBM model highlights potential important roles for immunomodulators that alter the properties of the T cell membrane such as cholesterol-derived and lipid-soluble compounds as well as changes in lipid composition in aging and pathological conditions which may pave the way for new therapies. A critical role for membrane bending in T cell activation suggests new designs for optimally engineering CAR-T cells and biTEs. Methods that measure piconewton forces and membrane bending on the nanometer scale can help assess the relevance of the TBM model.

**Reporting summary**. Further information on research design is available in the Nature Research Reporting Summary linked to this article.

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

## Acknowledgements

This work is supported by a grant from the Ministry of Science and Technology, Taipei, Taiwan (NSC99-2320-B-001-005-MY3). M.L.D. is supported by Kennedy Trust for Rheumatology Research. A.J. is supported by UKRI-BBSRC grant BB/M011224/1, Clarendon Fund (University of Oxford), and Oxford Interdisciplinary Bioscience DTP.

## Author contributions

M.A. and S.R. posited the main idea. M.A. wrote the original draft and prepared figures; M.A., A.J., M.D., and S.R. contributed in re-drafting and writing the paper; and substantial reviewing and correction was done by M.D. and S.R. All authors approved the final version of the paper for publication.

## Competing interests

The authors declare no competing interests.
