## [Peer Review File · Communications Biology]

Reviewers' comments:

Reviewer #1 (Remarks to the Author):

The manuscript by Al-Aghbar and colleagues gives a nice and comprehensive overview over potential models that consider mechanical forces to be imperative for T cell antigen recognition. I have only one criticism, which relates to the paper's exclusive focus on forces: many if not all experimental data could also be reconciled with models that do not include forces acting on the TCR at all. I suggest keeping a somewhat more open attitude to other models throughout the manuscript.

In addition, there are a few minor points which should be addressed:

- Several studies found evidence that tangential - but not normal - forces are involved in T cell triggering: Göhring et al. 2021, Nature Communications. 12:2502; Feng et al. 2017. PNAS 114:E8204-E8213. How does that fit into the models described here?
- Line 210: Since TCR triggering precedes LFA-1 activation, wouldn't forces transmitted via LFA-1 act too late?
- Line 290: Those data could as well be explained by the kinetic segregation model.
- Line 488: Piezo1 -> Piezo1
- Figure 8: the symbols for streptavidin and for the untriggered TCR are difficult to distinguish.

Reviewer #2 (Remarks to the Author):

General:

In the manuscript entitled: "Interplay between membrane topology and mechanical forces in TCR triggering and regulation of T cell activity", Al-Aghbar et al. proposed a model consisting of the following key components:

- "mechanical forces transmitted through engaged TCRs cause local bending of the T cell membrane on the nanometer scale..."
- The sources of mechanical forces on engaged TCRs include "surface receptor topology" and "Thermally-induced stochastic membrane fluctuations".
- Localized membrane bending triggers TCR "by changing the relative position of TCRs and associated signal transduction subunits", i.e. causing the dissociation of CD3 ITAMs from the plasma membrane due to "steric pressure increases".
- Overall, "the extent and duration of local membrane bending determines T cell signaling strength."

High-curvature membranes have been observed frequently at cytoskeleton protrusions, immunological synapses, and vesicles; however, the causative relationship between the membrane bending and TCR triggering has not been established. The authors proposed this "membrane-curvature-driving TCR mechanoactivation" model in an attempt to unify the kinetic segregation model and the "contradictory" close contact triggering evidence. The work is an extension of some of the authors' previous studies of the elongated-tethered BC3, OKT3 induced T cell activation without segregation, which suggested thermal membrane fluctuations as an alternative source of mechanical forces. Whereas the model provides an interesting speculation for the mechanism of TCR mechanosensing, this reviewer found insufficient support to some of the authors' bold claims. Evidence provided to relate mechanical forces and membrane bending to TCR triggering remains correlative. Due to the lack of further descriptions, validations, predictions, and outlooks, this reviewer cannot agree with the proposed model and cannot recommend acceptance of the manuscript in its current form.

Specific:

1) The authors stated membrane curvature as a general concept without specifying the scales. It seems to span from micrometers (as in the depiction of synapse) to nanometers (as in discussing how this change may result in TCR triggering). High membrane bending curvatures are often associated with structural proteins, heterogeneity of inner and outer leaflets, as well as raft

segregations, which require substantial energy input. The authors do not seem to appreciate how drastically different energy requirements would be for inducing membrane curvatures at these different scales. Instead of speculation, the authors should do the hard work to quantitatively estimate whether the proposed mechanisms (membrane thermal fluctuations and surface receptor topology) could afford the speculated curvature surrounding a single TCR-CD3 complex at nanometer scale and how big the passive force would be on engaged TCR should the nanoscale membrane bending exists. For example, following the Canham-Helfrich theory, one could estimate the thermally driven height variance to approximate $k_B T / q^4 / \kappa$. This indicates much smaller displacements over the size of TCR-CD3 complex and suggests that the crowding and steric pressure effects illustrated in Figure 4 are unlikely spontaneous. To the best of this reviewer's knowledge, there has been no evidence of membrane bending at the single TCR-CD3 level. Yet, the authors built their TCR triggering model on such a shaky ground with little evidence or rationale. It is appreciable though, membrane bending may come in play at larger scales before and after TCR triggering, such as TCR nanoclustering on microvilli tips on resting T cells and the formation of TCR microclusters and the development of synapse structures during T cell activation.

2) TCR ligation/triggering may further induce membrane curvature at the aforementioned scales, however, there's no evidence to support the notion that such curvature is a requirement of signaling initiation. The proposed effect of steric pressure increase caused by membrane curvature and molecular crowdedness were observed in artificial GUVs loaded with very high densities of molecules. The authors need to be cautious when translating these observations into a physiological mechanism, where the number of TCR-CD3 molecules are far below those in the artificial system. Unfortunately, the core component of the proposed model was built on such a translation. Without any other supporting evidence, this reviewer is not convinced that this is the mechanism of TCR mechanosensing, let alone the bold statement "the extent and duration of local membrane bending determines T cell signaling strength."

3) Consider two scenarios:

Scenario 1. Low curvature but high signaling- despite increasing membrane rigidity (10.1073/pnas.2004807117), cholesterol enhances T cell signaling (10.1038/nature17412). In addition, crowding effects dampen membrane fluctuations; however, the cooperativity of TCR has been reported (10.1126/scisignal.2000402).

Scenario 2. High curvature but low signaling- on tips of actin-driven protrusions, high curvature may disrupt the optimal TCR-coreceptor dimensional matching required for effective triggering (10.3389/fphy.2019.00045).

These scenarios make the correlation between membrane bending and TCR triggering ambiguous. The authors should provide explanation in order for their hypothesis of membrane-curvature-driving mechanoactivation not to be negated.

4) Whereas the authors' model provides another approach to visualize the impact of mechanical forces, this reviewer has concern about the low signal-to-noise ratio given its large intrinsic displacements in refs 78-81, which makes the model validation challenging. The authors should provide some measure for how well their approach works.

Overall, this reviewer found the model to be primitive and misleading due to lack of supporting evidence to its core components. However, the authors made very nice discussion on topics on TCR triggering as monomeric vs dimeric ligands, soluble vs immobilized ligands, membrane molecular topology, effect of CD45 and kinetic segregation models. Although some of these are hardly unified into their proposed membrane curvature model, this reviewer would encourage the authors to refine their model of membrane curvature, with the necessary caveats, and present these discussions to the field as a Perspective.

Reviewers' comments:

Reviewer #1 (Remarks to the Author):

The manuscript by Al-Aghbar and colleagues gives a nice and comprehensive overview over potential models that consider mechanical forces to be imperative for T cell antigen recognition. I have only one criticism, which relates to the paper's exclusive focus on forces: many if not all experimental data could also be reconciled with models that do not include forces acting on the TCR at all. I suggest keeping a somewhat more open attitude to other models throughout the manuscript.

We discussed throughout this perspective current explanations for TCR triggering, and we also tried to present a new viewpoint based on accumulating evidence of the role of force in TCR triggering. We tried to strike a balance in this perspective between current explanations of the published data and the possibility of further explanations. Indeed, we appreciate your kind suggestion to tone down the text when solid data is still missing. For example, we discussed the popular kinetic segregation (KS) model and how results using soluble monovalent ligands support this model (Section: 1. Soluble monovalent ligands do not activate T cells). We also discussed experiments conducted to test the model, current results and understanding in Section: 7. CD45 segregation is not required for TCR signaling. In the revised manuscript, we also added additional points regarding explanations from the cholesterol allosteric model in section 8: Membrane mechanosensing and mechanical stiffness potentiate TCR triggering.

However, the allosteric model of TCR regulation posits that cholesterol has affinity to TCR β and can bind only with non-triggered TCRs, inhibiting the overall TCR stimulation, and that cholesterol removal from around TCRs is required to initiate signaling [1]. Similarly, treating T cells with cholesterol sulfate, a natural derivative of membrane cholesterol, inhibits TCR avidity and disrupts TCR nanoclusters [2]. Yang et al. 2016 reported that inhibiting cholesterol esterification increases cholesterol levels in the membrane of CD8⁺ T cells and enhances their antitumor activity [3]. These diverse studies suggest that cholesterol may play a more complicated role in TCR triggering than simply adjusting membrane rigidity.

In addition, there are a few minor points which should be addressed:

- Several studies found evidence that tangential - but not normal - forces are involved in T cell triggering:

Göhring et al. 2021, Nature Communications. 12:2502;

Feng et al. 2017. PNAS 114:E8204-E8213.

How does that fit into the models described here?

Actually, both tangential as well as normal forces are involved in TCR triggering. Li et al. 2010 showed that both tangential (shear) forces as well as normal forces applied on TCRs induced Ca^{2+} flux in T cells (line 141-144 and figure 2.6.) [4]. Similarly, pushing or pulling TCRs using AFM induces Ca^{2+} flux [5].

Göhring et al. 2021, Nature Communications. 12:2502:

In this paper, the described forces are measured late after activation, and after immunological synapse formation. Tangential movement was driven by the internal machinery of the cell and powered by F-actin, which may lead to polarized delivery of ectosomes [6]. However, in our manuscript, we focus more about the initial triggering of TCRs.

Feng et al. 2017. PNAS 114:E8204-E8213:

This paper showed that tangential (shear) forces might more potently initiate triggering as compared to normal forces [7]. Based on Feng et al., the variations in TCR triggering can be explained by geometry, in which vertical tensile forces are equally distributed over ligated TCRs, which requires higher force input to induce TCR conformational changes and initiate triggering. By contrast, shear forces may pull TCR or TCR nanoclusters in one direction. Therefore, less force input is concentrated over fewer TCRs to induce triggering. In addition, vertical forces may pull TCRs away from the cytoskeleton beneath the plasma membrane as compared to shear forces, since the cytoskeleton is important to stabilize signaling, triggering by pulling forces may be lower.

We included this point in the revised manuscript on (section: **Mechanical force can initiate signaling through the TCR**):

The potency of tangential (shear) forces as compared to normal (pulling) forces was also described in detail by Feng et al. 2017 [7]. These results might be explained by geometry, in which vertical tensile forces are equally distributed over ligated TCRs, which requires higher force input to induce TCR conformational changes and initiate triggering. By contrast, shear forces may pull TCRs or TCR nanoclusters in one direction. Therefore, less force input is required over fewer TCRs to induce triggering. In addition, vertical forces may pull TCRs away from the cytoskeleton beneath the plasma membrane to a greater extent than shear forces, which may negatively impact triggering during vertical pulling since the cytoskeleton acts as a scaffold for signaling complexes [7].

• Line 210: Since TCR triggering precedes LFA-1 activation, wouldn't forces transmitted via LFA-1 act too late?

No, it is not too late. When using natural pMHC complexes on SLBs, T cells cannot efficiently respond without ICAM-1, supporting the notion that LFA-1/ICAM-1 interactions are required to establish an environment in which TCR-pMHC interactions can take place and function. Furthermore, Saito's group showed that micro-adhesion rings surrounding TCR microclusters formed very early in TCR triggering (~30 sec) and moved along with TCR clusters to the cSMAC. Disruption of micro-adhesion ring formation by downmodulating its components led to impaired T cell adhesion and TCR triggering [8]. Therefore, we speculate that LFA-1/ICAM-1 binding stabilizes TCR/pMHC interactions, and that outside-in signaling of LFA-1/ICAM-1 is not delayed from the initial signal TCRs receive. The role of LFA-1/ICAM-1 is likely to enable TCR-pMHC interactions and to intensify signaling initiated by TCR ligation, possibly by increasing membrane bending around TCRs immediately upon engagement.

• Line 290: Those data could as well be explained by the kinetic segregation model.

Yes, that is right, and we edited the text to reflect this. The KS model also fits to the observation that CD45 segregates from engaged TCRs bound by tethered ligands, but it fails to explain other results such as the finding that CD45 can co-localize with activated TCR [9], and data showing that CD45 segregation follows TCR triggering by phase separation [10]. In addition, we argue that most of the experiments that were designed to prove the KS model by using truncated or chimeric CD45 were also affecting CD45 localization relative to Lck in lipid domains or reducing CD45 catalytic activity [11; 12].

The manuscript was revised on (section **1. Soluble monovalent ligands do not activate T cells**)

These results are consistent with both the KS model and with deformation of the T cell membrane by physical forces. In fact, the KS model is very popular because of the obvious segregation of CD45 from the immune synapse and TCR microclusters, and studies showing that disruption of CD45 segregation abolished signaling [13; 14; 15]. However, other studies show that CD45 segregation is not required for TCR triggering [9; 10; 16].

(Discussed in detail in (section: **7. CD45 segregation is not required for TCR signaling**)

- Line 488: Peizo1 -> Piezo1

Corrected

- Figure 8: the symbols for streptavidin and for the untriggered TCR are difficult to distinguish. Thanks, streptavidin shape and color are now modified, and the whole figure is slightly modified to show the effect of both membrane curvature and CD45 segregation in TCR signaling.

Reviewer #2 (Remarks to the Author):

General:

In the manuscript entitled: “Interplay between membrane topology and mechanical forces in TCR triggering and regulation of T cell activity”, Al-Aghbar et al. proposed a model consisting of the following key components:

- “mechanical forces transmitted through engaged TCRs cause local bending of the T cell membrane on the nanometer scale...”
- The sources of mechanical forces on engaged TCRs include “surface receptor topology” and “Thermally-induced stochastic membrane fluctuations”.
- Localized membrane bending triggers TCR “by changing the relative position of TCRs and associated signal transduction subunits”, i.e. causing the dissociation of CD3 ITAMs from the plasma membrane due to “steric pressure increases”.
- Overall, “the extent and duration of local membrane bending determines T cell signaling strength.”

High-curvature membranes have been observed frequently at cytoskeleton protrusions, immunological synapses, and vesicles; however, the causative relationship between the membrane bending and TCR triggering has not been established. The authors proposed this “membrane-curvature-driving TCR mechanoactivation” model in an attempt to unify the kinetic segregation model and the "contradictory" close contact triggering evidence. The work is an extension of some of the authors' previous studies of the elongated-tethered BC3, OKT3 induced T cell activation without segregation, which suggested thermal membrane fluctuations as an alternative source of mechanical forces. Whereas the model provides an interesting speculation for the mechanism of TCR mechanosensing, this reviewer found insufficient support to some of the authors' bold claims. Evidence provided to relate mechanical forces and membrane bending to TCR triggering remains correlative. Due to the lack of further descriptions,

validations, predictions, and outlooks, this reviewer cannot agree with the proposed model and cannot recommend acceptance of the manuscript in its current form.

Specific:

1) The authors stated membrane curvature as a general concept without specifying the scales. It seems to span from micrometers (as in the depiction of synapse) to nanometers (as in discussing how this change may result in TCR triggering).

We are referring to nanometer scale changes, not the micrometer scale changes. We stated that in different places in the text (please refer to the abstract)

High membrane bending curvatures are often associated with structural proteins, heterogeneity of inner and outer leaflets, as well as raft segregations, which require substantial energy input. The authors do not seem to appreciate how drastically different energy requirements would be for inducing membrane curvatures at these different scales. Instead of speculation, the authors should do the hard work to quantitatively estimate whether the proposed mechanisms (membrane thermal fluctuations and surface receptor topology) could afford the speculated curvature surrounding a single TCR-CD3 complex at nanometer scale and how big the passive force would be on engaged TCR should the nanoscale membrane bending exists.

For example, following the Canham-Helfrich theory, one could estimate the thermally driven height variance to approximate $k_B T/q^4/\kappa$. This indicates much smaller displacements over the size of TCR-CD3 complex and suggests that the crowding and steric pressure effects illustrated in Figure 4 are unlikely spontaneous. To the best of this reviewer's knowledge, there has been no evidence of membrane bending at the single TCR-CD3 level. Yet, the authors built their TCR triggering model on such a shaky ground with little evidence or rationale. It is appreciable though, membrane bending may come in play at larger scales before and after TCR triggering, such as TCR nanoclustering on microvilli tips on resting T cells and the formation of TCR microclusters and the development of synapse structures during T cell activation.

As reviewed by Chabanon et al., there are 5 biological mechanisms by which cellular membranes are deformed: changes in lipid composition that causes lipid asymmetry in inner and outer membrane leaflets, scaffolding of some proteins on one side of the membrane, protein oligomerization or clustering, hydrophobic insertion of proteins with amphipathic helices, and protein crowding [17].

In fact, it is widely accepted that when two membranes from two adjacent cells come into contact, the insulating glycocalyxes are sorted, cause conformational changes, or drive membrane bending to allow close contact of short ligands/receptors such as pMHC/TCR interaction. This was proposed since decades by Timothy Springer (1990) [18]. This was also suggested by Burroughs and Wulfig (2002) [19]. In addition, membrane bending is resulted from close pMHC/TCR contact, and these binding energy drives protein sorting and membrane relaxation to more stable situation, as discussed by James and Vale (2012) [20].

In our model, we propose that membrane bending results from direct input of force from membrane receptor topology as well as membrane fluctuations. Is energy input from membrane topology sufficient to induce membrane bending: We think it is sufficient for the following reasons:

1. Our membrane bending model looks similar to the elegant system created by Kim 2020 [21], in which live cells with biotinylated membrane lipids were moved by interactions with magnetic-avidin-coated beads via optical tweezers to generate membrane curvature. Similar experiments were performed on biotinylated lipid vesicles bound to an avidin coated glass surface to generate membrane vertical movement. Kim estimated that applying 9 pN of force on vesicles with a radius in a deformation-free configuration ($r_{vr} = 70$ nm) and with radius of the area of contact with the underlying surface ($r_{vc} = 58$ nm), the membrane extended up to ~ 60 nm. [21]. Actually, the intercellular space between engaged APCs and T cells is less than 60 nm, which means that 9 pN of input is sufficient to induce the estimated membrane bending. But is 9 pN sufficient to induce triggering? This was answered by Liu et al. in 2016, in which a tension map to induce TCR triggering was prepared, and according to Liu et al., applying a force of 12 pN to individual TCRs can initiate TCR triggering [22]. In other words, applying a force of 12 pN to individual TCR causes TCR triggering and is more than required to bend T cell membrane.

2. Bending rigidity values obtained from many studies at fixed ligand concentration showed that both T cells and APCs have membrane bending rigidity of $25k_B T$ [23].
3. Applying Kim 2020 to live systems, E-cadherin was pulled from the surface of human bone osteosarcoma epithelial cells, and the displacement of the membrane around pulled E-cadherin was measured. Results collected from E-cadherin or lipid vesicles showed that the biological membrane can be always deformed (displaced) when applying forces to surface proteins. The displacement depends on the size of the cells or vesicles [21].
4. Increasing membrane rigidity by treating cells with excessive amount of cholesterol (20 mM) while pulling E-cadherins changed the curve of pulling and relaxation of cadherin, indicating that membrane lipid deformation is dependent on membrane rigidity (more force is required for membrane bending (deformation)) [21].
5. In one interesting paper, Hatzakis et al. developed a method to measure curvature in deformed membranes based on the inserted wedges of alkyl motifs of membrane curved proteins such as the N-terminus of endophilin A1 (eAH) [24]. This system was based on the curves generated by liposomes, which could be estimated using different variants such as liposome radius, total area, and lipid thickness. In this model, it was found that higher deformed membranes (like that in 50 nm liposomes) may generate higher binding sites for these wedges. Due to topological variations in membrane proteins, a high degree of membrane curvature is expected to be formed around engaged TCRs surrounded with dispersed free or bound proteins such as CD45 and LFA-1, respectively. Therefore, if membrane deformation occurs, curvature sensing proteins with alkyl motifs should be detected around individual TCRs or TCR nanoclusters. In this regard, endophilin A2, a closely related endophilin protein that regulates membrane bending through a BAR domain [25], colocalizes with TCRs and pZap70 upon TCR triggering, and it plays an important role in TCR internalization. A deficiency of this protein significantly reduces T cell activation [26].

Figure 4 does not illustrate how mechanical forces are provided by membrane topology, but rather that biological membranes fluctuate over time (stochastic membrane fluctuations). Stochastic membrane fluctuations have been described in many previous publications [27; 28; 29; 30].

We summarized these articles on section (Role of membrane bending in T cell activation) to support the notion that the T cell membrane can be deformed when applying forces on individual TCRs or TCR nanoclusters.

One of the biggest challenges is to prove that membrane bending at the nanometer scale can be achieved by molecular topology of the T cell membrane. In an elegant study performed by Kim and colleagues, beads coupled to liposomes or cells by biotin-streptavidin or cadherin interactions, were pulled by optical tweezers to generate membrane curvature, in which the liposome radius, surface area of contact, temperature, the applied forces, and membrane displacement were measured. Kim estimated that applying 9 pN of force on vesicles with a 70 nm radius in a deformation-free configuration and with 58 nm radius of the area of contact with the underlying surface caused the membrane to extend up to ~ 60 nm [21]. Although a liposome is obviously different from a cell, the intercellular distance between engaged APCs and T cells needs to be less than 15 nm, suggesting that 9 pN of force may be sufficient to induce membrane bending at engaged TCRs. In an elegant study, Liu applied forces to individual TCRs to generate a tension map showing that about 12 pN of force acting on individual TCRs can initiate TCR signaling [22]. Taken together, these studies indicate that the amount of force needed to generate membrane deformation is similar to the force required for TCR triggering. Hatzakis et al. developed an interesting method to measure curvature in deformed membranes based on insertion of wedges of alkyl motifs of membrane curved proteins such as the N-terminus of endophilin A1 (eAH); A higher degree of membrane deformation creates more binding sites for these wedges [24]. Interestingly, endophilin A2, a closely related endophilin protein that regulates membrane bending through a BAR domain [25], colocalizes with TCR microclusters and pZap70 upon TCR triggering. This protein also plays an important role in TCR internalization, and its deficiency reduces T cell activation significantly [26].

2) TCR ligation/triggering may further induce membrane curvature at the aforementioned scales, however, there's no evidence to support the notion that such curvature is a requirement of signaling initiation. The proposed effect of steric pressure increase caused by membrane curvature and

molecular crowdedness were observed in artificial GUVs loaded with very high densities of molecules. The authors need to be cautious when translating these observations into a physiological mechanism, where the number of TCR-CD3 molecules are far below those in the artificial system. Unfortunately, the core component of the proposed model was built on such a translation. Without any other supporting evidence, this reviewer is not convinced that this is the mechanism of TCR mechanosensing, let alone the bold statement “the extent and duration of local membrane bending determines T cell signaling strength.”

This model is correlative, based on observations listed in the manuscript and further examples listed above. We based this statement “the extent and duration of local membrane bending determines T cell signaling strength) on the fact that catch bonds last longer than slip bonds, but we agree that this statement was not accurate, so we modified it to:

. In this “TCR Bending Mechanosignal” model (TBM), the necessary extent and duration of local membrane bending is achieved by agonist that form catch bonds, which may reduce the noise signal coming from non-specific binding since the slip bonds ruptures before phosphorylation that prevents relaxation of TCR ITAMs to the resting state. We compare other proposed models to the proposed new standpoint in data obtained from accumulative publications in this field.

3) Consider two scenarios:
Scenario 1. Low curvature but high signaling- despite increasing membrane rigidity (10.1073/pnas.2004807117), cholesterol enhances T cell signaling (10.1038/nature17412). In addition, crowding effects dampen membrane fluctuations; however, the cooperativity of TCR has been reported (10.1126/scisignal.2000402).
Scenario 2. High curvature but low signaling- on tips of actin-driven protrusions, high curvature may disrupt the optimal TCR-coreceptor dimensional matching required for effective triggering (10.3389/fphy.2019.00045).

These scenarios make the correlation between membrane bending and TCR triggering ambiguous. The authors should provide explanation in order for their hypothesis of membrane-curvature-

driving mechanoactivation not to be negated.

Scenario 1

Cholesterol increases membrane thickness and lipid packing which results in more bending rigidity [31]. In the reference you mentioned, Yang et al. 2016 used a model system where cholesterol esterification was inhibited. They showed that cholesterol level increased in CD8+ but not CD4+ T cell membranes, which enhanced TCR clustering and immune synapse formation, and as a result higher anticancer activity [3].

By contrast, in our model, it is expected that more force input is required to bend cholesterol rich membrane as compared to normal membranes. Or in other words, cholesterol may inhibit TCR signaling because membrane bending can't be achieved at low force levels. However, other studies that are not consistent with Yang et al. and show inhibitory effects of the presence of cholesterol on TCR triggering:

1. Treating human T cell lines or naïve human T cells with cholesterol impairs calcium signaling and T cell activation by anti-CD3/anti-CD28 stimulation [32].
2. Accumulation of cholesterol in aged mouse T cells decreases their potency to be activated [33; 34].
3. High cholesterol levels in aged human T cells decreases membrane fluidity leading to impaired lipid rafts and reduced T cell activation and proliferation [35].
4. Cholesterol sulfate, a natural derivative of membrane cholesterol, inhibits CD3 ITAM phosphorylation. There is a tendency of cholesterol and cholesterol sulfate to bind TCR β , and treating T cells with cholesterol sulfate inhibits TCR avidity and disrupts TCR nanoclusters [2].
5. Schamel proposed a cholesterol allosteric model of TCR regulation. The model states that the TCR is controlled by conformational switches depending on their engagement to pMHC, and that cholesterol levels in the T cell membrane helps maintain TCRs in a resting conformation. Three findings contribute to the cholesterol allosteric model: i. Cholesterol can bind only with non-triggered but not with triggered TCRs. ii. Cholesterol binding inhibits TCR triggering. iii. Cholesterol removal is associated with TCR triggering [1].

These diverse studies suggest that cholesterol may play a more complicated role in TCR triggering than adjusting membrane rigidity.

We added the following to the text to clarify the relationship between cholesterol and membrane bending (Section: **8. Membrane mechanosensing and mechanical stiffness potentiate TCR triggering**):

It is therefore expected that increasing membrane rigidity may require more force to generate membrane bending. Cholesterol, for example, increases membrane rigidity and negatively impacts TCR triggering. Treating human T cell lines or naïve human T cells with cholesterol upon activation with anti-CD3/anti-CD28 impairs calcium signaling and T cell activation [32]. Accumulation of cholesterol in aged mouse T cells also correlates with decreased activation potential [33; 34]. Similarly, high cholesterol levels were detected in aged human T cells. At high cholesterol levels, membrane fluidity decreases leading to impaired lipid raft formation and reduced T cell activation and proliferation [35]. However, the allosteric model of TCR regulation posits that cholesterol has affinity to TCR β and can bind only with non-triggered TCRs, inhibiting the overall TCR stimulation, and that cholesterol removal from around TCRs is required to initiate signaling [1]. Similarly, treating T cells with cholesterol sulfate, a natural derivative of membrane cholesterol, inhibits TCR avidity and disrupts TCR nanoclusters [2]. Yang et al. 2016 reported that inhibiting cholesterol esterification increases cholesterol levels in the membrane of CD8⁺ T cells and enhances their antitumor activity [3]. These diverse studies suggest that cholesterol may play a more complicated role in TCR triggering than simply adjusting membrane rigidity.

Scenario 2

Based on Sage et al. 2012, invadosomes like protrusions (ILPs) average ~430 nm in depth and ~350 nm in width [36]. This curved projection is very large as compared to individual TCRs or TCR nanoclusters, which means that the membrane bending in the tip of ILP is still less than that around an engaged TCR surrounded by LFA-1/ICAM-1 and CD45. However, it is expected that

invadosomes provide sites for signaling and help overcome the glycocalyx barrier to form close intercellular contacts with APCs to enhance antigen probing efficiency [36].

We added the following to section **(8. Membrane mechanosensing and mechanical stiffness potentiate TCR triggering)**:

ILPs have average size of ~430 nm in depth and ~350 nm in width that help overcome the glycocalyx barrier and form close intercellular contacts with APCs to enhance antigen probing efficiency [36]. These protrusions are stabilized by calcium mobilization, which may support signal amplification [36].

4) Whereas the authors' model provides another approach to visualize the impact of mechanical forces, this reviewer has concern about the low signal-to-noise ratio given its large intrinsic displacements in refs 78-81, which makes the model validation challenging. The authors should provide some measure for how well their approach works.

These references describe the natural membrane fluctuations that appear with and without TCR ligation. We expect that the following happens in regard to reduce the noise signal (signal from antagonist):

1. Ligation to pMHC is specific, binding to weak antagonists will be with lower affinity as compared to agonist ligands.
2. As described elsewhere, catch bonds are formed only by agonists. Membrane fluctuations are necessary to distinguish catch bonds from slip bonds, now most of the noise signals (slip bonds formed with antagonists) are removed.
3. Initial triggering starts with exposed ITAMs. Catch bonds may induce cooperative binding with other TCRs, calcium flux stabilizes invadosome protrusions, and within seconds, LFA-1 inside-out signal is activated.
4. The force provided by LFA-1/ICAM-1 interactions may intensify bending around TCRs, and that is required for signal amplification. If there are still low-affinity bound TCRs, bonds are disrupted.

The following was added to the summary:

We expect that TCR triggering is regulated tightly by membrane bending to reduce signaling noise.

Therefore, the following are expected to happen sequentially to start TCR signaling:

1. T cells meet APCs that present an agonist peptide on MHC molecules.
2. T cell uses microvilli/ILPs to sense the availability of agonist pMHC and to overcome the glycocalyx barrier.
3. Catch bonds are formed only by agonists whereas membrane fluctuations rapidly break slip bonds to reduce signaling noise (from antagonist).
4. Stabilized catch bonds may induce cooperative binding with other TCRs.
5. The topological variations between TCRs and other membrane proteins induce membrane bending, which results in further pressure to break non-specific binding. Conformational changes within the TCR complex components relative to the T cell membrane cause dissociation of ζ chain from the inner leaflet of the plasma membrane.
6. Initial triggering starts, phosphorylated ζ induces downstream signaling that stabilizes the structure of ILPs by calcium-dependent mechanisms. In addition, talin is recruited to induce LFA-1 activation to ligate to ICAM-1 on the APC surface.
7. LFA-1/ICAM-1 interactions induce outside-in signaling that increases LFA-1 binding affinity. LFA-1/ICAM-1 binding intensifies membrane bending to amplify TRC signaling.
8. Microadhesion rings move in arrays to the center of T cell/APC contact area forming the cSMAC, while other microadhesion rings keep appearing at the pSMAC and moving to the cSMAC powered by the cell cytoskeleton.

Overall, this reviewer found the model to be primitive and misleading due to lack of supporting evidence to its core components. However, the authors made very nice discussion on topics on TCR triggering as monomeric vs dimeric ligands, soluble vs immobilized ligands, membrane molecular topology, effect of CD45 and kinetic segregation models. Although some of these are hardly unified into their proposed membrane curvature model, this reviewer would encourage the authors to refine their model of membrane curvature, with the necessary caveats, and present these

discussions to the field as a Perspective.

** See the Nature Portfolio author and referees' website at www.nature.com/authors for information about policies, services and author benefits

Communications Biology is committed to improving transparency in authorship. As part of our efforts in this direction, we are now requesting that all authors identified as 'corresponding author' create and link their Open Researcher and Contributor Identifier (ORCID) with their account on the Manuscript Tracking System prior to acceptance. ORCID helps the scientific community achieve unambiguous attribution of all scholarly contributions. You can create and link your ORCID from the home page of the Manuscript Tracking System by clicking on 'Modify my Springer Nature account' and following the instructions in the link below. Please also inform all co-authors that they can add their ORCID to their accounts and that they must do so prior to acceptance. <https://www.springernature.com/gp/researchers/orcid/orcid-for-nature-research>

If you experience problems in linking your ORCID, please contact the Platform Support Helpdesk.

Our flexible approach during the COVID-19 pandemic
If you need more time at any stage of the peer-review process, please do let us know. While

our systems will continue to remind you of the original timelines, we aim to be as flexible as possible during the current pandemic.

COMMSBIO - This email has been sent through the Springer Nature Tracking System NY-610A-NPG&MTS

Confidentiality

Statement:

This e-mail is confidential and subject to copyright. Any unauthorised use or disclosure of its contents is prohibited. If you have received this email in error please notify our Manuscript Tracking System Helpdesk team at <http://platformsupport.nature.com>.

Details of the confidentiality and pre-publicity policy may be found here <http://www.nature.com/authors/policies/confidentiality.html>

Privacy Policy | Update Profile

References:

- [1] M. Swamy, K. Beck-Garcia, E. Beck-Garcia, F.A. Hartl, A. Morath, O.S. Yousefi, E.P. Dopfer, E. Molnar, A.K. Schulze, R. Blanco, A. Borroto, N. Martin-Blanco, B. Alarcon, T. Hofer, S. Minguet, and W.W. Schamel, A Cholesterol-Based Allosteric Model of T Cell Receptor Phosphorylation. *Immunity* 44 (2016) 1091-101.
- [2] F. Wang, K. Beck-Garcia, C. Zorzin, W.W. Schamel, and M.M. Davis, Inhibition of T cell receptor signaling by cholesterol sulfate, a naturally occurring derivative of membrane cholesterol. *Nat Immunol* 17 (2016) 844-50.
- [3] W. Yang, Y. Bai, Y. Xiong, J. Zhang, S. Chen, X. Zheng, X. Meng, L. Li, J. Wang, C. Xu, C. Yan, L. Wang, C.C. Chang, T.Y. Chang, T. Zhang, P. Zhou, B.L. Song, W. Liu, S.C. Sun, X. Liu, B.L. Li, and C. Xu, Potentiating the antitumour response of CD8(+) T cells by modulating cholesterol metabolism. *Nature* 531 (2016) 651-5.
- [4] Y.C. Li, B.M. Chen, P.C. Wu, T.L. Cheng, L.S. Kao, M.H. Tao, A. Lieber, and S.R. Roffler, Cutting Edge: mechanical forces acting on T cells immobilized via the TCR complex can trigger TCR signaling. *J Immunol* 184 (2010) 5959-63.
- [5] K.H. Hu, and M.J. Butte, T cell activation requires force generation. *The Journal of Cell Biology* 213 (2016) 535-542.
- [6] J. Gohring, F. Kellner, L. Schrangl, R. Platzer, E. Klotzsch, H. Stockinger, J.B. Huppa, and G.J. Schutz, Temporal analysis of T-cell receptor-imposed forces via quantitative single molecule FRET measurements. *Nat Commun* 12 (2021) 2502.

- [7] Y. Feng, K.N. Brazin, E. Kobayashi, R.J. Mallis, E.L. Reinherz, and M.J. Lang, Mechanosensing drives acuity of alphabeta T-cell recognition. *Proc Natl Acad Sci U S A* 114 (2017) E8204-E8213.
- [8] A. Hashimoto-Tane, M. Sakuma, H. Ike, T. Yokosuka, Y. Kimura, O. Ohara, and T. Saito, Micro-adhesion rings surrounding TCR microclusters are essential for T cell activation. *J Exp Med* 213 (2016) 1609-25.
- [9] M.A. Al-Aghbar, Y.S. Chu, B.M. Chen, and S.R. Roffler, High-Affinity Ligands Can Trigger T Cell Receptor Signaling Without CD45 Segregation. *Front Immunol* 9 (2018) 713.
- [10] X. Su, J.A. Ditlev, E. Hui, W. Xing, S. Banjade, J. Okrut, D.S. King, J. Taunton, M.K. Rosen, and R.D. Vale, Phase separation of signaling molecules promotes T cell receptor signal transduction. *Science* 352 (2016) 595-9.
- [11] C. Irles, A. Symons, F. Michel, T.R. Bakker, P.A. van der Merwe, and O. Acuto, CD45 ectodomain controls interaction with GEMs and Lck activity for optimal TCR signaling. *Nat Immunol* 4 (2003) 189-97.
- [12] D.M. Desai, J. Sap, J. Schlessinger, and A. Weiss, Ligand-mediated negative regulation of a chimeric transmembrane receptor tyrosine phosphatase. *Cell* 73 (1993) 541-54.
- [13] K. Choudhuri, M. Parker, A. Milicic, D.K. Cole, M.K. Shaw, A.K. Sewell, G. Stewart-Jones, T. Dong, K.G. Gould, and P.A. van der Merwe, Peptide-major histocompatibility complex dimensions control proximal kinase-phosphatase balance during T cell activation. *J Biol Chem* 284 (2009) 26096-105.
- [14] S.P. Cordoba, K. Choudhuri, H. Zhang, M. Bridge, A.B. Basat, M.L. Dustin, and P.A. van der Merwe, The large ectodomains of CD45 and CD148 regulate their segregation from and inhibition of ligated T-cell receptor. *Blood* 121 (2013) 4295-302.
- [15] K. Choudhuri, D. Wiseman, M.H. Brown, K. Gould, and P.A. van der Merwe, T-cell receptor triggering is critically dependent on the dimensions of its peptide-MHC ligand. *Nature* 436 (2005) 578-82.
- [16] H. Cai, J. Muller, D. Depoil, V. Mayya, M.P. Sheetz, M.L. Dustin, and S.J. Wind, Full control of ligand positioning reveals spatial thresholds for T cell receptor triggering. *Nat Nanotechnol* 13 (2018) 610-617.
- [17] M. Chabanon, J.C. Stachowiak, and P. Rangamani, Systems biology of cellular membranes: a convergence with biophysics. *Wiley Interdiscip Rev Syst Biol Med* 9 (2017).
- [18] T.A. Springer, Adhesion receptors of the immune system. *Nature* 346 (1990) 425-34.
- [19] N.J. Burroughs, and C. Wulfig, Differential segregation in a cell-cell contact interface: the dynamics of the immunological synapse. *Biophys J* 83 (2002) 1784-96.
- [20] J.R. James, and R.D. Vale, Biophysical mechanism of T-cell receptor triggering in a reconstituted system. *Nature* 487 (2012) 64-9.
- [21] J. Kim, Probing nanomechanical responses of cell membranes. *Sci Rep* 10 (2020) 2301.
- [22] Y. Liu, L. Blanchfield, V.P.-Y. Ma, R. Andargachew, K. Galior, Z. Liu, B. Evavold, and K. Salaita, DNA-based nanoparticle tension sensors reveal that T-cell receptors transmit defined pN forces to their antigens for enhanced fidelity. *Proceedings of the National Academy of Sciences* 113 (2016) 5610-5615.
- [23] L. Li, J. Hu, B. Rzycki, X. Wang, H. Wu, and F. Song, Influence of lipid rafts on pattern formation during T-cell adhesion. *New Journal of Physics* 23 (2021).
- [24] N.S. Hatzakis, V.K. Bhatia, J. Larsen, K.L. Madsen, P.Y. Bolinger, A.H. Kunding, J. Castillo, U. Gether, P. Hedegard, and D. Stamou, How curved membranes recruit amphipathic helices and protein anchoring motifs. *Nat Chem Biol* 5 (2009) 835-41.
- [25] J. Bai, Z. Hu, J.S. Dittman, E.C. Pym, and J.M. Kaplan, Endophilin functions as a membrane-bending molecule and is delivered to endocytic zones by exocytosis. *Cell* 143 (2010) 430-41.
- [26] U. Norin, C. Rintisch, L. Meng, F. Forster, D. Ekman, J. Tuncel, K. Klocke, J. Backlund, M. Yang, M.Y. Bonner, G.F. Lahore, J. James, K. Shchetynsky, M. Bergquist, I. Gjertsson, N. Hubner, L. Backdahl,

- and R. Holmdahl, Endophilin A2 deficiency protects rodents from autoimmune arthritis by modulating T cell activation. *Nat Commun* 12 (2021) 610.
- [27] A.Y. Krol, M. Grinfeldt, S. Levin, and A. Smilgavichus, Local mechanical oscillations of the cell surface within the range 0.2–30 Hz. *European Biophysics Journal* 19 (1990) 93-99.
- [28] C. Monzel, D. Schmidt, C. Kleusch, D. Kirchenbuchler, U. Seifert, A.S. Smith, K. Sengupta, and R. Merkel, Measuring fast stochastic displacements of bio-membranes with dynamic optical displacement spectroscopy. *Nat Commun* 6 (2015) 8162.
- [29] C. Monzel, D. Schmidt, U. Seifert, A.S. Smith, R. Merkel, and K. Sengupta, Nanometric thermal fluctuations of weakly confined biomembranes measured with microsecond time-resolution. *Soft Matter* 12 (2016) 4755-68.
- [30] H. Turlier, D.A. Fedosov, B. Audoly, T. Auth, N.S. Gov, C. Sykes, J.F. Joanny, G. Gompper, and T. Betz, Equilibrium physics breakdown reveals the active nature of red blood cell flickering. *Nature Physics* 12 (2016) 513-519.
- [31] S. Chakraborty, M. Doktorova, T.R. Molugu, F.A. Heberle, H.L. Scott, B. Dzikovski, M. Nagao, L.R. Stingaciu, R.F. Standaert, F.N. Barrera, J. Katsaras, G. Khelashvili, M.F. Brown, and R. Ashkar, How cholesterol stiffens unsaturated lipid membranes. *Proc Natl Acad Sci U S A* 117 (2020) 21896-21905.
- [32] D.H. Nguyen, J.C. Espinoza, and D.D. Taub, Cellular cholesterol enrichment impairs T cell activation and chemotaxis. *Mech Ageing Dev* 125 (2004) 641-50.
- [33] A. Tamir, M.D. Eisenbraun, G.G. Garcia, and R.A. Miller, Age-dependent alterations in the assembly of signal transduction complexes at the site of T cell/APC interaction. *J Immunol* 165 (2000) 1243-51.
- [34] G.G. Garcia, and R.A. Miller, Single-cell analyses reveal two defects in peptide-specific activation of naive T cells from aged mice. *J Immunol* 166 (2001) 3151-7.
- [35] A. Larbi, N. Douziech, G. Dupuis, A. Khalil, H. Pelletier, K.P. Guerard, and T. Fulop, Jr., Age-associated alterations in the recruitment of signal-transduction proteins to lipid rafts in human T lymphocytes. *J Leukoc Biol* 75 (2004) 373-81.
- [36] P.T. Sage, L.M. Varghese, R. Martinelli, T.E. Sciuto, M. Kamei, A.M. Dvorak, T.A. Springer, A.H. Sharpe, and C.V. Carman, Antigen recognition is facilitated by invadosome-like protrusions formed by memory/effector T cells. *J Immunol* 188 (2012) 3686-99.

REVIEWERS' COMMENTS:

Reviewer #1 (Remarks to the Author):

The authors addressed my comments adequately. I recommend publication

Reviewer #2 (Remarks to the Author):

Overall the manuscript has been much improved. The authors added a clear summary of their model and removed/softened the over-statements. Instead of claiming membrane bending initiates signaling, they now presented it as ways to facilitate noise reduction (filtering out weak (slip) bonds) and ITAM exposure. They also added missing literature and discussion to address reviewers' concerns. These in general provide good support to their model.